# Guaranteed Recovery of One-Hidden-Layer Neural Networks via Cross Entropy

## Abstract

We study model recovery for data classification, where the training labels are generated from a one-hidden-layer fully-connected neural network with sigmoid activations, and the goal is to recover the weight vectors of the neural network. We prove that under Gaussian inputs, the empirical risk function using cross entropy exhibits strong convexity and smoothness *uniformly* in a local neighborhood of the ground truth, as soon as the sample complexity is sufficiently large. This implies that if initialized in this neighborhood, which can be achieved via the tensor method, gradient descent converges linearly to a critical point that is provably close to the ground truth without requiring a fresh set of samples at each iteration. To the best of our knowledge, this is the first global convergence guarantee established for the empirical risk minimization using cross entropy via gradient descent for learning one-hidden-layer neural networks, at the near-optimal sample and computational complexity with respect to the network input dimension.

## 1 Introduction

Neural networks have attracted a significant amount of research interest in recent years due to the success of deep neural networks (LeCun et al., 2015) in practical domains such as computer vision and artificial intelligence (Russakovsky et al., 2015; He et al., 2016; Silver et al., 2016). However, the theoretical underpinnings behind such success remains mysterious to a large extent. Efforts have been taken to understand which classes of functions can be represented by deep neural networks (Cybenko, 1989; Hornik et al., 1989; Barron, 1993; Telgarsky, 2016), when (stochastic) gradient descent is effective for optimizing a non-convex loss function (Dauphin et al., 2014), and why these networks generalize well (Zhang et al., 2016; Bartlett et al., 2017; Brutzkus et al., 2017).

One important line of research that has attracted extensive attention is a model-recovery setup, i.e., given that the training samples $(\boldsymbol{x}_i, y_i) \sim (\boldsymbol{x}, y)$ are generated i.i.d. from a distribution $\mathcal{D}$ based on a neural network model with the ground truth parameter $\boldsymbol{W}^\star$, the goal is to recover the underlying model parameter $\boldsymbol{W}^\star$, which is important for the network to generalize well (Mondelli & Montanari, 2018). Previous studies along this topic can be mainly divided into two types of data generations. First, a regression problem, for example, assumes that each sample $y$ is generated as $y = \frac{1}{K}\sum_{k=1}^{K} \phi(\boldsymbol{w}_k^{\star\top}\boldsymbol{x})$, where $\boldsymbol{w}_k \in \mathbb{R}^d$ is the weight vector of the $k$th neuron, $1 \leq k \leq K$, and the input $\boldsymbol{x} \in \mathbb{R}^d$ is Gaussian. This type of regression problem has been studied in various settings. In particular, (Soltanolkotabi, 2017) studied the single-neuron model under ReLU activation, (Zhong et al., 2017b) studied the one-hidden-layer multi-neuron network model, and (Li & Yuan, 2017) studied a two-layer feedforward networks with ReLU activations and identity mapping. Second, for a classification problem, suppose each label $y \in \{0, 1\}$ is drawn under the conditional distribution $\mathbb{P}(y = 1|\boldsymbol{x}) = \frac{1}{K}\sum_{k=1}^{K} \phi(\boldsymbol{w}_k^{\star\top}\boldsymbol{x})$, where $\boldsymbol{w}_k^\star \in \mathbb{R}^d$ is the weight vector of the $k$th neuron, $1 \leq k \leq K$, and the input $\boldsymbol{x} \in \mathbb{R}^d$ is Gaussian. Such a problem has been studied in (Mei et al., 2016) in the case with a single neuron.

For both the regression and the classification settings, in order to recover the neural network parameters, all previous studies considered (stochastic) gradient descent over the squared loss, i.e.,

$$\ell_{qu}\left(\boldsymbol{W}; \boldsymbol{x}, y\right) = \frac{1}{2}\left(y - \frac{1}{K}\sum_{i=1}^{K}\phi\left(\boldsymbol{w}_i^\top \boldsymbol{x}\right)\right)^2, \tag{1}$$

which yields gradient and Hessian in relatively simple forms to assist the landscape characterization of the function as well as model recovery analysis.

However, for the classification problem, the cross entropy objective used in practice takes the following form

$$\ell\left(\boldsymbol{W};\boldsymbol{x},y\right) = -y \cdot \log\left(\frac{1}{K}\sum_{i=1}^{K}\phi\left(\boldsymbol{w}_i^\top \boldsymbol{x}\right)\right) - (1-y)\cdot\log\left(1 - \frac{1}{K}\sum_{i=1}^{K}\phi\left(\boldsymbol{w}_i^\top \boldsymbol{x}\right)\right). \quad (2)$$

The geometry as well as the model recovery problem based on the entropy loss function have not yet been understood. It is expected that such a loss function is very challenging to analyze, not just because it is nonconvex with multiple neurons, but also because the gradient and Hessian take much more complicated forms compared with the squared loss. The main focus of this paper is to develop technical analysis for guaranteed model recovery under the challenging cross entropy loss function in eq. (2) for the classification problem in the multi-neuron case.

Furthermore, previous studies provided two types of statistical guarantees for such model recovery problems using the squared loss. More specifically, (Zhong et al., 2017b) showed that in the local neighborhood of the ground truth, the Hessian of the *empirical* loss function is positive definite for each *given* point under *independent* high probability event. Hence, their guarantee for gradient descent to converge to the ground truth requires a *fresh* set of samples at every iteration, thus the total sample complexity will depend on the number of iterations. On the other hand, studies such as (Mei et al., 2016; Soltanolkotabi, 2017) establish certain types of *uniform geometry* such as strong convexity so that resampling per iteration is not needed for gradient descent to have guaranteed linear convergence as long as it enters such a local neighborhood. However, such a stronger statistical guarantee *without per-iteration resampling* have only been shown for the squared loss function. In this paper, we aim at developing such a strong statistical guarantee for the loss function in eq. (2), which is much more challenging but more practical than the squared loss for the classification problem.

## 1.1 OUR CONTRIBUTIONS

This study provides the first performance guarantee for the recovery of one-hidden-layer neural networks using the *cross entropy* loss function, to the best of our knowledge. More specifically, our contributions are summarized as follows.

- For multi-neuron classification problem with sigmoid activations, we show that, if the input is Gaussian, the empirical risk function $f_n(\boldsymbol{W}) = \frac{1}{n}\sum_{i=1}^{n}\ell\left(\boldsymbol{W};\boldsymbol{x}_i\right)$ based on the cross entropy loss in eq. (2) is *uniformly* strongly convex in a local neighborhood of the ground truth $\boldsymbol{W}^\star$ of size $O(1/K^{3/2})$ as soon as the sample size is $O(dK^5\log^2 d)$, where $d$ is the input dimension and $K$ is the number of neurons.

- We further show that, if initialized in this neighborhood, gradient descent converges linearly to a critical point $\widehat{\boldsymbol{W}}_n$ (which we show to exist), with a sample complexity of $O(dK^5\log^2 d)$, which is near-optimal up to a polynomial factor in $K$ and $\log d$. Due to the nature of quantized labels here, the recover of $\boldsymbol{W}^\star$ is only up to certain statistical accuracy, and $\widehat{\boldsymbol{W}}_n$ converges to $\boldsymbol{W}^\star$ at a rate of $O(\sqrt{dK^{9/2}\log n/n})$ in the Frobenius norm. Furthermore, such a convergence guarantee *does not* require a fresh set of samples at each iteration due to the *uniform* strong convexity in the local neighborhood. To obtain $\epsilon$-accuracy, it requires a computational complexity of $O(ndK^2\log(1/\epsilon))$.

- We adopt the tensor method proposed in (Zhong et al., 2017b), and show it provably provides an initialization in the neighborhood of the ground truth. In particular, our proof replaces the homogeneous assumption on activation functions in (Zhong et al., 2017b) by a mild condition on the curvature of activation functions around $\boldsymbol{W}^\star$, which holds for a larger class of activation functions including *sigmoid* and *tanh*.

In order to analyze the challenging cross-entropy loss function, our proof develops various new machineries in order to exploit the statistical information of the geometric curvatures, including the gradient and Hessian of the empirical risk, and to develop covering arguments to guarantee uniform concentrations. Our technique also yields similar performance guarantees for the classification problem using the squared loss in eq. (1), which we omit due to space limitations, as it is easier to analyze than cross entropy.

## 1.2 RELATED WORK

Due to page limitations we focus on the most relevant literature on theoretical and algorithmic aspects of learning shallow neural networks via nonconvex optimization.

The parameter recovery viewpoint is relevant to the success of non-convex learning in signal processing problems such as matrix completion, phase retrieval, blind deconvolution, dictionary learning and tensor decomposition (Sun & Luo, 2016; Candès et al., 2015; Ge & Ma, 2017; Ge et al., 2016; Sun et al., 2015; Bhojanapalli et al., 2016; Ma et al., 2017), to name a few. The statistical model for data generation effectively removes worst-case instances and allows us to focus on average-case performance, which often possess much benign geometric properties that enable global convergence of simple local search algorithms.

The studies of one-hidden-layer network model can be further categorized into two classes, landscape analysis and model recovery. In the landscape analysis, it is known that if the network size is large enough compared to the data input, then there are no spurious local minima in the optimization landscape, and all local minima are global (Soltanolkotabi et al., 2017; Boob & Lan, 2017; Safran & Shamir, 2016; Nguyen & Hein, 2017). For the case with multiple neurons ($2 \leq K \leq d$) in the under-parameterized setting, the work of Tian (Tian, 2017) studied the landscape of the population squared loss surface with ReLU activations. In particular, there exist spurious bad local minima in the optimization landscape (Ge et al., 2017; Safran & Shamir, 2017) even at the population level. Zhong et. al. (Zhong et al., 2017b) provided several important characterizations for the local Hessian for the regression problem for a variety of activation functions for the squared loss.

In the model recovery problem, the number of neurons is smaller than the dimension of inputs. In the case with a single neuron ($K = 1$), under Gaussian input, (Soltanolkotabi, 2017) showed that gradient descent converges linearly when the activation function is ReLU, i.e. $\phi(z) = \max\{z, 0\}$, with a zero initialization, as long as the sample complexity is $O(d)$ for the regression problem. On the other end, (Mei et al., 2016) showed that when $\phi(\cdot)$ has bounded first, second and third derivatives, there is no other critical points than the unique global minimum (within a constrained region of interest), and (projected) gradient descent converges linearly with an arbitrary initialization, as long as the sample complexity is $O(d \log^2 d)$ with sub-Gaussian inputs for the classification problem using the squared loss. Moreover, (Zhong et al., 2017b) shows that the ground truth From a technical perspective, our study differs from all the aforementioned work in that the cross entropy loss function we analyze has a very different form. Furthermore, we study the model recovery classification problem under the multi-neuron case, which has not been studied before.

Finally, we note that several papers study one-hidden-layer or two-layer neural networks with different structures under Gaussian input. For example, (Brutzkus & Globerson, 2017; Du et al., 2017a;b; Zhong et al., 2017a) studied the non-overlapping convolutional neural network, (Li & Yuan, 2017) studied a two-layer feedforward networks with ReLU activations and identity mapping, and (Feizi et al., 2017) introduced the Porcupine Neural Network. These results are not directly comparable to ours since both the networks and the loss functions are different.

## 1.3 PAPER ORGANIZATION AND NOTATIONS

The rest of the paper is organized as follows. Section 2 describes the problem formulation. Section 3 presents the main results on local geometry and local linear convergence of gradient descent. Section 4 discusses the initialization method. Numerical examples are demonstrated in Section 5, and finally, conclusions are drawn in Section 6.

Throughout this paper, we use boldface letters to denote vectors and matrices, e.g. $\boldsymbol{w}$ and $\boldsymbol{W}$. The transpose of $\boldsymbol{W}$ is denoted by $\boldsymbol{W}^{\top}$, and $\|\boldsymbol{W}\|$, $\|\boldsymbol{W}\|_{\mathrm{F}}$ denote the spectral norm and the Frobenius norm. For a positive semidefinite (PSD) matrix $\boldsymbol{A}$, we write $\boldsymbol{A} \succeq 0$. The identity matrix is denoted by $\boldsymbol{I}$. The gradient and the Hessian of a function $f(\boldsymbol{W})$ is denoted by $\nabla f(\boldsymbol{W})$ and $\nabla^2 f(\boldsymbol{W})$, respectively. Let $\sigma_i(\boldsymbol{W})$ denote the $i$-th singular value of $\boldsymbol{W}$. Denote $\|\cdot\|_{\psi_1}$ as the sub-exponential norm of a random variable. We use $c, C, C_1, \ldots$ to denote constants whose values may vary from line to line. For nonnegative functions $f(x)$ and $g(x)$, $f(x) = O(g(x))$ means there exist positive constants $c$ and $a$ such that $f(x) \leq cg(x)$ for all $x \geq a$; $f(x) = \Omega(g(x))$ means there exist positive constants $c$ and $a$ such that $f(x) \geq cg(x)$ for all $x \geq a$.

## 2 PROBLEM FORMULATION

We first describe the generative model for training data, and then describe the gradient descent algorithm for learning the network weights.

### 2.1 MODEL

Suppose we are given $n$ training samples $\{(\boldsymbol{x}_i, y_i)\}_{i=1}^n \sim (\boldsymbol{x}, y)$ that are drawn i.i.d., where $\boldsymbol{x} \sim \mathcal{N}(\boldsymbol{0}, \boldsymbol{I})$. Assume the activation function is sigmoid, i.e. $\phi(z) = 1/(1 + e^{-z})$ for all $z$. Conditioned on $\boldsymbol{x} \in \mathbb{R}^d$, we consider the classification setting, where $y$ is mapped to a discrete label using the one-hidden layer neural network model as follows:

$$\mathbb{P}(y = 1 | \boldsymbol{x}) = \frac{1}{K} \sum_{k=1}^K \phi(\boldsymbol{w}_k^{\star\top} \boldsymbol{x}). \tag{3}$$

and $\mathbb{P}(y = 0 | \boldsymbol{x}) = 1 - \mathbb{P}(y = 1 | \boldsymbol{x})$, where $K$ is the number of neurons.

Our goal is to estimate $\boldsymbol{W}^\star = [\boldsymbol{w}_1^\star, \cdots, \boldsymbol{w}_K^\star]$, via minimizing the following empirical risk function:

$$f_n(\boldsymbol{W}) = \frac{1}{n} \sum_{i=1}^n \ell(\boldsymbol{W}; \boldsymbol{x}_i), \tag{4}$$

where $\ell(\boldsymbol{W}; \boldsymbol{x}) := \ell(\boldsymbol{W}; \boldsymbol{x}, y)$ is the cross entropy loss, i.e., the negative log-likelihood function, i.e.,

$$\ell(\boldsymbol{W}; \boldsymbol{x}) = -\left[ y \cdot \log\left( \frac{1}{K} \sum_{i=1}^K \phi\left(\boldsymbol{w}_i^\top \boldsymbol{x}\right) \right) + (1 - y) \cdot \log\left( 1 - \frac{1}{K} \sum_{i=1}^K \phi\left(\boldsymbol{w}_i^\top \boldsymbol{x}\right) \right) \right].$$

Let $\boldsymbol{w} = \text{vec}(\boldsymbol{W}) = \left[\boldsymbol{w}_1^\top, \cdots, \boldsymbol{w}_K^\top\right]^\top \in \mathbb{R}^{dK}$ be the vectorized form of $\boldsymbol{W}$. With slight abuse of notation, we denote the gradient and Hessian of $\ell(\boldsymbol{W}; \boldsymbol{x})$ with respect to the vector $\boldsymbol{w}$.

### 2.2 GRADIENT DESCENT

To estimate $\boldsymbol{W}^\star$, since (4) is a highly nonconvex function, vanilla gradient descent with an arbitrary initialization may get stuck at local minima. Therefore, we implement the gradient descent algorithm with a well-designed initialization scheme that is described in detail in Section 4. The update rule is given as

$$\boldsymbol{W}_{t+1} = \boldsymbol{W}_t - \eta \nabla f_n(\boldsymbol{W}_t),$$

where $\eta$ is the step size. The algorithm is summarized in Algorithm 1.

---

**Algorithm 1** Gradient Descent

---

**Input**: Training data $\{(\boldsymbol{x}_i, y_i)\}_{i=1}^n$, step size $\eta$, iteration $T$
**Initialization**: $\boldsymbol{W}_0 \leftarrow \text{INITIALIZATION}\left(\{(\boldsymbol{x}_i, y_i)\}_{i=1}^n\right)$
**Gradient Descent**: for $t = 0, 1, \cdots, T$

$$\boldsymbol{W}_{t+1} = \boldsymbol{W}_t - \eta \nabla f_n(\boldsymbol{W}_t).$$

**Output**: $\boldsymbol{W}_T$

---

We note that throughout the execution of the algorithm, the same set of training samples is used which is the standard implementation of gradient descent. This is in sharp contrast to existing work such as Zhong et al. (2017b) that employs the impractical scheme of *resampling*, where a *fresh* set of training samples is used at every iteration of gradient descent.

## 3 MAIN RESULTS

Before stating our main results, we first introduce an important quantity regarding $\phi(z)$ that captures the geometric properties of the loss function, distilled in (Zhong et al., 2017b).

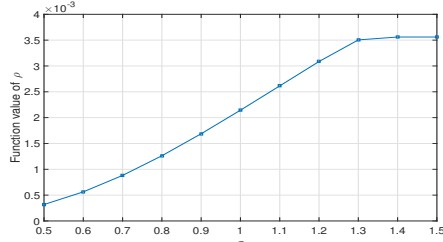

Figure 1: $\rho(\sigma)$ for sigmoid activation.

**Definition 1.** *Let* $\alpha_q(\sigma) = \mathbb{E}_{z \sim \mathcal{N}(0,1)}[\phi'(\sigma \cdot z)z^q], \forall q \in \{0,1,2\}$, *and* $\beta_q(\sigma) = \mathbb{E}_{z \sim \mathcal{N}(0,1)}[\phi'^2(\sigma \cdot z)z^q], \forall q \in \{0,2\}$. *Define* $\rho(\sigma)$ *as*

$$\rho(\sigma) = \min\{\beta_0(\sigma) - \alpha_0^2(\sigma) - \alpha_1^2(\sigma),$$
$$\beta_2(\sigma) - \alpha_1^2(\sigma) - \alpha_2^2(\sigma)\}.$$

Note that the definition here is different from that in (Zhong et al., 2017b, Property 3.2) but consistent with (Zhong et al., 2017b, Lemma D.4) which removes the third term in (Zhong et al., 2017b, Property 3.2). For the activation function considered in this paper, the first two terms suffice. We depict $\rho(\sigma)$ as a function of $\sigma$ in a certain range for the sigmoid activation in Fig. 1. It is easy to observe that $\rho(\sigma) > 0$ for all $\sigma > 0$.

### 3.1 LOCAL STRONG CONVEXITY

We first characterize the local strong convexity of $f_n(\boldsymbol{W})$ in a neighborhood of the ground truth $\boldsymbol{W}^\star$. Let $\mathbb{B}(\boldsymbol{W}^\star, r)$ denote a Euclidean ball centered at $\boldsymbol{W}^\star \in \mathbb{R}^{d \times K}$ with a radius $r$, i.e.

$$\mathbb{B}(\boldsymbol{W}^\star, r) = \left\{ \boldsymbol{W} \in \mathbb{R}^{d \times K} : \|\boldsymbol{W} - \boldsymbol{W}^\star\|_{\mathrm{F}} \leq r \right\}.$$

Let $\sigma_i := \sigma_i(\boldsymbol{W}^\star)$ denote the $i$-th singular value of $\boldsymbol{W}^\star$. Let the condition number be $\kappa = \sigma_1/\sigma_K$, and $\lambda = \prod_{i=1}^K (\sigma_i/\sigma_K)$. The following theorem guarantees the Hessian of the empirical risk function $f_n(\boldsymbol{W})$ in the local neighborhood of $\boldsymbol{W}^\star$ is positive definite with high probability.

**Theorem 1.** *For the classification model* (3) *with sigmoid activation function, assume* $\|\boldsymbol{W}^\star\|_{\mathrm{F}} \leq 1$, *then there exists some constant* $C$, *such that if*

$$n \geq C \cdot dK^5 \log^2 d \cdot \left( \frac{\kappa^2 \lambda}{\rho(\sigma_K)} \right)^2,$$

*then with probability at least* $1 - d^{-10}$, *for all* $\boldsymbol{W} \in \mathbb{B}(\boldsymbol{W}^\star, r)$,

$$\Omega\left( \frac{1}{K^2} \cdot \frac{\rho(\sigma_K)}{\kappa^2 \lambda} \right) \cdot \boldsymbol{I} \preceq \nabla^2 f_n(\boldsymbol{W}) \preceq C \cdot \boldsymbol{I}$$

*hold, where* $r := \min\left\{ \frac{C}{K^{\frac{3}{2}}} \cdot \frac{\rho(\sigma_K)}{\kappa^2 \lambda}, 0.7 \right\}$.

We note that all column permutations of $\boldsymbol{W}^\star$ are equivalent global minima of the loss function, and Theorem 1 applies to all such permutation matrices of $\boldsymbol{W}^\star$. The proof of Theorem 1 is outlined in Appendix A. Theorem 1 guarantees that the Hessian of the empirical cross-entropy loss function $f_n(\boldsymbol{W})$ is positive definite (PD) in a neighborhood of the ground truth $\boldsymbol{W}^\star$, as long as $\rho(\sigma_K) > 0$ (i.e. $\boldsymbol{W}^\star$ is full-column rank), when the sample size $n$ is sufficiently large for the sigmoid activation. The bounds in Theorem 1 depend on the dimension parameters of the network ($n$ and $K$), as well as the activation function and the ground truth ($\rho(\sigma_K)$, $\lambda$). As a special case, suppose $\boldsymbol{W}^\star$ is composed of orthonormal columns with $\rho(\sigma_K) = O(1)$, $\kappa = 1$, $\lambda = 1$. Then, Theorem 1 guarantees $\Omega(1/K^2)\boldsymbol{I} \preceq \nabla^2 f_n(\boldsymbol{W}) \preceq C$ within the neighborhood $\mathbb{B}(\boldsymbol{W}^\star, \Omega(1/K\sqrt{K}))$, as soon as the sample complexity $n = \Omega(dK^5 \log^2 d)$. The sample complexity is order-wise near-optimal in $d$ up to polynomial factors of $K$ and $\log d$, since the number of unknown parameters is $dK$.

### 3.2 Performance Guarantees of Gradient Descent

For the classification problem, due to the nature of quantized labels, $W^\star$ is no longer a critical point of $f_n(W)$. By the strong convexity of the empirical risk function $f_n(W)$ in the local neighborhood of $W^\star$, there can exist at most one critical point in $\mathbb{B}(W^\star, r)$, which is the unique local minimizer in $\mathbb{B}(W^\star, r)$ if it exists. The following theorem shows that there indeed exists such a critical point $\widehat{W}_n$, which is provably close to the ground truth $W^\star$, and gradient descent converges linearly to $\widehat{W}_n$.

**Theorem 2.** *For the classification model* (3) *with sigmoid activation function, and assume* $\|W^\star\|_F \leq 1$, *there exist some constants* $C, C_1 > 0$ *such that if the sample size* $n \geq C \cdot dK^5 \log^2 d \cdot \left( \frac{\kappa^2 \lambda}{\rho(\sigma_K)} \right)^2$, *then with probability at least* $1 - d^{-10}$, *there exists a unique critical point* $\widehat{W}_n$ *in* $\mathbb{B}(W^\star, r)$ *with* $r := \min\left\{ \frac{c}{K^{3/2}} \cdot \frac{\rho(\sigma_K)}{\kappa^2 \lambda}, 0.7 \right\}$, *which satisfies*

$$\left\| \widehat{W}_n - W^\star \right\|_F \leq C_1 \frac{K^{9/4} \kappa^2 \lambda}{\rho(\sigma_K)} \sqrt{\frac{d \log n}{n}}. \tag{5}$$

*Moreover, if the initial point* $W_0 \in \mathbb{B}(W^\star, r)$, *then gradient descent converges linearly to* $\widehat{W}_n$, *i.e.*

$$\left\| W_t - \widehat{W}_n \right\|_F \leq (1 - H_{\min} \eta)^t \left\| W_0 - \widehat{W}_n \right\|_F \tag{6}$$

*where* $H_{\min} = \Omega\left( \frac{1}{K^2} \cdot \frac{\rho(\sigma_K)}{\kappa^2 \lambda} \right)$, *as long as the step size* $\eta = \Omega\left( \frac{1}{K^2} \cdot \frac{\rho(\sigma_K)}{\kappa^2 \lambda} \right)$.

Similarly to Theorem 1, Theorem 2 also holds for all column permutations of $W^\star$. The proof can be found in Appendix B. Theorem 2 guarantees that there exists a critical point $\widehat{W}_n$ in $\mathbb{B}(W^\star, r)$ which converges to $W^\star$ at the rate of $O(K^{9/4} \sqrt{d \log n/n})$, and therefore $W^\star$ can be recovered consistently as $n$ goes to infinity. Moreover, gradient descent converges linearly to $\widehat{W}_n$ at a linear rate, as long as it is initialized in the basin of attraction. To achieve $\epsilon$-accuracy, i.e. $\left\| W_t - \widehat{W}_n \right\|_F \leq \epsilon$, it requires a computational complexity of $O\left( ndK^2 \log(1/\epsilon) \right)$, which is linear in $n$, $d$ and $\log(1/\epsilon)$.

## 4 Initialization

Our initialization adopts the tensor method proposed in (Zhong et al., 2017b). In this section, we first briefly describe this method, and then present the performance guarantee of the initialization with remarks on the differences from that in (Zhong et al., 2017b).

### 4.1 Preliminary and Algorithm

This subsection briefly introduces the tensor method proposed in (Zhong et al., 2017b), to which a reader can refer for more details. We first define a product $\widetilde{\otimes}$ as follows. If $v \in \mathbb{R}^d$ is a vector and $I$ is the identity matrix, then $v \widetilde{\otimes} I = \sum_{j=1}^d [v \otimes e_j \otimes e_j + e_j \otimes v \otimes e_j + e_j \otimes e_j \otimes v]$. If $M$ is a symmetric rank-$r$ matrix factorized as $M = \sum_{i=1}^r s_i v_i v_i^\top$ and $I$ is the identity matrix, then

$$M \widetilde{\otimes} I = \sum_{i=1}^r s_i \sum_{j=1}^d \sum_{l=1}^6 A_{l,i,j},$$

where $A_{1,i,j} = v_i \otimes v_i \otimes e_j \otimes e_j$, $A_{2,i,j} = v_i \otimes e_j \otimes v_i \otimes e_j$, $A_{3,i,j} = e_j \otimes v_i \otimes v_i \otimes e_j$, $A_{4,i,j} = v_i \otimes e_j \otimes e_j \otimes v_i$, $A_{5,i,j} = e_j \otimes v_i \otimes e_j \otimes v_i$ and $A_{6,i,j} = e_j \otimes e_j \otimes v_i \otimes v_i$.

**Definition 2.** *Define* $M_1$, $M_2$, $M_3$, $M_4$ *and* $m_{1,i}$, $m_{2,i}$, $m_{3,i}$, $m_{4,i}$ *as follows:*
$M_1 = \mathbb{E}[y \cdot x]$,
$M_2 = \mathbb{E}[y \cdot (x \otimes x - I)]$,
$M_3 = \mathbb{E}[y \cdot (x^{\otimes 3} - x \widetilde{\otimes} I)]$,
$M_4 = \mathbb{E}[y \cdot (x^{\otimes 4} - (x \otimes x) \widetilde{\otimes} I + I \widetilde{\otimes} I)]$,
$m_{1,i} = \gamma_1(\|w_i^\star\|)$,
$m_{2,i} = \gamma_2(\|w_i^\star\|) - \gamma_0(\|w_i^\star\|)$,
$m_{3,i} = \gamma_3(\|w_i^\star\|) - 3\gamma_1(\|w_i^\star\|)$,
$m_{4,i} = \gamma_4(\|w_i^\star\|) + 3\gamma_0(\|w_i^\star\|) - 6\gamma_2(\|w_i^\star\|)$,
*where* $\gamma_j(\sigma) = \mathbb{E}_{z \sim \mathcal{N}(0,1)}[\phi(\sigma \cdot z)z^j]$, $\forall j = 0, 1, 2, 3, 4$.

**Definition 3.** *Let $\boldsymbol{\alpha} \in \mathbb{R}^d$ denote a randomly picked vector. We define $\boldsymbol{P}_2$ and $\boldsymbol{P}_3$ as follows: $\boldsymbol{P}_2 = \boldsymbol{M}_{j_2}(\boldsymbol{I}, \boldsymbol{I}, \boldsymbol{\alpha}, \cdots, \boldsymbol{\alpha})$,[1] where $j_2 = \min\{j \geq 2 | \boldsymbol{M}_j \neq 0\}$, and $\boldsymbol{P}_3 = \boldsymbol{M}_{j_3}(\boldsymbol{I}, \boldsymbol{I}, \boldsymbol{I}, \boldsymbol{\alpha}, \cdots, \boldsymbol{\alpha})$, where $j_3 = \min\{j \geq 3 | \boldsymbol{M}_j \neq 0\}$.*

We further denote $\overline{\boldsymbol{w}} = \boldsymbol{w}/\|\boldsymbol{w}\|$. The initialization algorithm based on the tensor method is summarized in Algorithm 2, which includes two major steps. Step 1 first estimates the direction of each column of $\boldsymbol{W}^\star$ by decomposing $\boldsymbol{P}_2$ to approximate the subspace spanned by $\{\overline{\boldsymbol{w}}_1^\star, \overline{\boldsymbol{w}}_2^\star, \cdots, \overline{\boldsymbol{w}}_K^\star\}$ (denoted by $\boldsymbol{V}$), then reduces the third-order tensor $P_3$ to a lower-dimension tensor $\boldsymbol{R}_3 = \boldsymbol{P}_3(\boldsymbol{V}, \boldsymbol{V}, \boldsymbol{V}) \in \mathbb{R}^{K \times K \times K}$, and applys non-orthogonal tensor decomposition on $\boldsymbol{R}_3$ to output the estimate $s_i \boldsymbol{V}^\top \overline{\boldsymbol{w}}_i^\star$, where $s_i \in \{1, -1\}$ is a random sign. Step 2 approximates the magnitude of $\boldsymbol{w}_i^\star$ and the sign $s_i$ by solving a linear system of equations.

---

**Algorithm 2** Initialization via Tensor Method

---

**Input:** Partition $n$ pairs of data $\{(\boldsymbol{x}_i, y_i)\}_{i=1}^n$ into three parts $\mathcal{D}_1, \mathcal{D}_2, \mathcal{D}_3$.
**Output:**
1: Estimate $\widehat{\boldsymbol{P}}_2$ of $\boldsymbol{P}_2$ from data set $\mathcal{D}_1$.
2: $\boldsymbol{V} \leftarrow \text{POWERMETHOD}(\widehat{\boldsymbol{P}}_2, K)$.
3: Estimate $\widehat{\boldsymbol{R}}_3$ of $\boldsymbol{P}_3(\boldsymbol{V}, \boldsymbol{V}, \boldsymbol{V})$ from data set $\mathcal{D}_2$.
4: $\{\widehat{\boldsymbol{u}}_i\}_{i \in [K]} \leftarrow \text{KCL}(\widehat{\boldsymbol{R}}_3)$.
5: $\{\boldsymbol{w}_i^{(0)}\}_{i \in [K]} \leftarrow \text{RECMAG}(\boldsymbol{V}, \{\widehat{\boldsymbol{u}}_i\}_{i \in [K]}, \mathcal{D}_3)$.

---

## 4.2 PERFORMANCE GUARANTEE OF INITIALIZATION

For the classification problem, we make the following technical assumptions, similarly in (Zhong et al., 2017b, Assumption 5.3) for the regression problem.

**Assumption 1.** *The activation function $\phi(z)$ satisfies the following conditions:1. If $M_j \neq 0$, then*

$$\sum_{i=1}^K m_{j,i} \left(\boldsymbol{w}_i^{\star\top} \boldsymbol{\alpha}\right)^{j-2} \overline{\boldsymbol{w}}_i^\star \overline{\boldsymbol{w}}_i^{\star\top} \neq \boldsymbol{0} \quad \forall j,$$

$$\sum_{i=1}^K m_{j,i} \left(\overline{\boldsymbol{w}}_i^{\star\top} \boldsymbol{\alpha}\right)^{j-3} (\boldsymbol{V}^\top \overline{\boldsymbol{w}}_i^\star) \text{vec}((\boldsymbol{V}^\top \overline{\boldsymbol{w}}_i^\star)(\boldsymbol{V}^\top \overline{\boldsymbol{w}}_i^\star)^\top)^\top \neq 0 \quad for \quad j \geq 3$$

*2. At least one of $M_3$ and $M_4$ is non-zero.*

Furthermore, we do not require the homogeneous assumption ((i.e., $\phi(az) = a^p z$ for an integer $p$)) required in (Zhong et al., 2017b), which can be restrictive. Instead, we assume the following condition on the curvature of the activation function around the ground truth, which holds for a larger class of activation functions such as sigmoid and tanh.

**Assumption 2.** *Let $l_1$ be the index of the first nonzero $M_i$ where $i = 1, \ldots, 4$. For the activation function $\phi(\cdot)$, there exists a positive constant $\delta$ such that $m_{l_1,i}(\cdot)$ is strictly monotone over the interval $(\|\boldsymbol{w}_i^\star\| - \delta, \|\boldsymbol{w}_i^\star\| + \delta)$, and the derivative of $m_{l_1,i}(\cdot)$ is lower bounded by some constant for all $i$.*

We next present the performance guarantee for the initialization algorithm in the following theorem.

**Theorem 3.** *For the classification model (3), under Assumptions 1 and 2, if the sample size $n \geq d\text{poly}(K, \kappa, t, \log d, 1/\epsilon)$, then the output $\boldsymbol{W}_0 \in \mathbb{R}^{d \times K}$ of Algorithm 2 satisfies*

$$\|\boldsymbol{W}_0 - \boldsymbol{W}^\star\|_{\mathrm{F}} \leq \epsilon \text{poly}(K, \kappa) \|\boldsymbol{W}^\star\|_{\mathrm{F}}, \tag{7}$$

*with probability at least $1 - d^{-\Omega(t)}$.*

The proof of Theorem 3 consists of (a) showing the estimation of the direction of $\boldsymbol{W}^\star$ is sufficiently accurate and (b) showing the approximation of the norm of $\boldsymbol{W}^\star$ is accurate enough. Our proof of part (a) is the same as that in (Zhong et al., 2017b), but our argument in part (b) is different, where we relax the homogeneous assumption on activation functions. More details can be found in the supplementary materials in Appendix C.

---

[1]See (15) in the supplemental materials for definition.

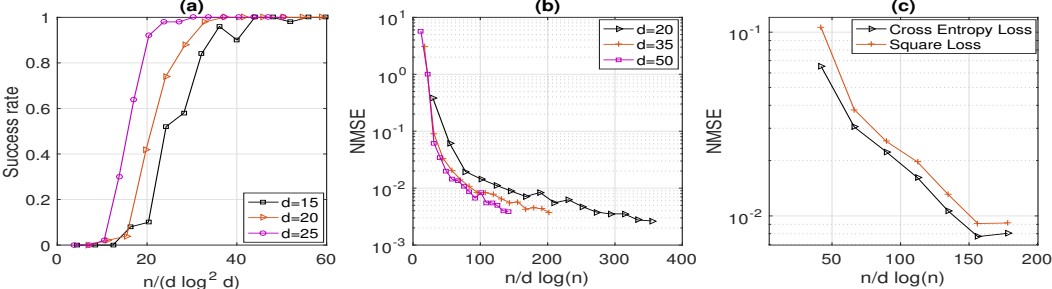

Figure 2: Fix $K = 3$. (a) Success rate of converging to the same local minima with respect to the sample complexity for various $d$; (b) Average estimation error of gradient descent in a local neighborhood of the ground truth with respect to the sample complexity for various $d$; (c) Average estimation error of gradient descent using different objective functions in a local neighborhood of the ground truth with respect to the sample complexity when $d = 20$.

## 5 NUMERICAL EXPERIMENTS

In this section, we first implement gradient descent to verify that the empirical risk function is strongly convex in the local region around $\boldsymbol{W}^\star$. If we initialize multiple times in such a local region, it is expected that gradient descent converges to the same critical point $\widehat{\boldsymbol{W}}_n$, with the same set of training samples. Given a set of training samples, we randomly initialize multiple times, and then calculate the variance of the output of gradient descent. Denote the output of the $\ell$th run as $\widehat{\boldsymbol{w}}_n^{(\ell)} = \mathrm{vec}(\widehat{\boldsymbol{W}}_n^{(\ell)})$ and the mean of the runs as $\bar{\boldsymbol{w}}$. The error is calculated as $\mathrm{SD}_n = \sqrt{\frac{1}{L}\sum_{\ell=1}^{L} \|\widehat{\boldsymbol{w}}_n^{(\ell)} - \bar{\boldsymbol{w}}\|^2}$, where $L = 20$ is the total number of random initializations. Adopted in (Mei et al., 2016), it quantifies the standard deviation of the estimator $\widehat{\boldsymbol{W}}_n$ under different initializations with the same set of training samples. We say an experiment is successful, if $\mathrm{SD}_n \leq 10^{-2}$.

Figure 2 (a) shows the successful rate of gradient descent by averaging over 50 sets of training samples for each pair of $n$ and $d$, where $K = 3$ and $d = 15, 20, 25$ respectively. The maximum iterations for gradient descent is set as $\mathrm{iter}_{\max} = 3500$. It can be seen that as long as the sample complexity is large enough, gradient descent converges to the same local minima with high probability.

We next show that the statistical accuracy of the local minimizer for gradient descent if it is initialized close enough to the ground truth. Suppose we initialize around the ground truth such that $\|\boldsymbol{W}_0 - \boldsymbol{W}_\star\|_{\mathrm{F}} \leq 0.1 \cdot \|\boldsymbol{W}_\star\|_{\mathrm{F}}$. We calculate the average estimation error as $\sum_{\ell=1}^{L} \|\widehat{\boldsymbol{W}}_n^{(\ell)} - \boldsymbol{W}^\star\|_{\mathrm{F}}^2 / (L\|\boldsymbol{W}^\star\|_{\mathrm{F}}^2)$ over $L = 100$ Monte Carlo simulations with random initializations. Fig. 2 (b) shows the average estimation error with respect to the sample complexity when $K = 3$ and $d = 20, 35, 50$ respectively. It can be seen that the estimation error decreases gracefully as we increase the sample size and matches with the theoretical prediction of error rates reasonably well.

We further compare the performance of gradient descent algorithm applied to both the cross entropy loss and the squared loss, respectively. As shown in Fig 2 (c), when $K = 3$, $d = 20$, cross entropy loss with gradient descent achieves a much lower error than the squared loss. Clearly, the cross entropy loss is favored in the classification problem over the squared loss.

## 6 CONCLUSIONS

In this paper, we have studied the model recovery of a one-hidden-layer neural network using the cross entropy loss in a multi-neuron classification problem. In particular, we have characterized the sample complexity to guarantee local strong convexity in a neighborhood (whose size we have characterized as well) of the ground truth when the training data are generated from a classification model. This guarantees that with high probability, gradient descent converges linearly to the ground truth if initialized properly. In the future, it will be interesting to extend the analysis in this paper to more general class of activation functions, particularly ReLU-like activations; and more general network structures, such as convolutional neural networks (Du et al., 2017b; Zhong et al., 2017a).

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

## A  PROOF OF THEOREM 1

To begin, denote the population loss function as

$$f(\boldsymbol{W}) = \mathbb{E}\left[f_n(\boldsymbol{W})\right] = \mathbb{E}\left[\ell\left(\boldsymbol{W}; \boldsymbol{x}\right)\right], \tag{8}$$

where the expectation is taken with respect to the distribution of the training sample $(\boldsymbol{x}; y)$.

The proof of Theorem 1 follows the following steps:

1. We first show that the Hessian $\nabla^2 f(\boldsymbol{W})$ of the population loss function is smooth with respect to $\nabla^2 f(\boldsymbol{W}^\star)$ (Lemma 1);

2. We then show that $\nabla^2 f(\boldsymbol{W})$ satisfies local strong convexity and smoothness in a neighborhood of $\boldsymbol{W}^\star$, $\mathbb{B}(\boldsymbol{W}^\star, r)$ with appropriately chosen radius by leveraging similar properties of $\nabla^2 f(\boldsymbol{W}^\star)$ (Lemma 2);

3. Next, we show that the Hessian of the empirical loss function $\nabla^2 f_n(\boldsymbol{W})$ is close to its popular counterpart $\nabla^2 f(\boldsymbol{W})$ uniformly in $\mathbb{B}(\boldsymbol{W}^\star, r)$ with high probability (Lemma 3).

4. Finally, putting all the arguments together, we establish $\nabla^2 f_n(\boldsymbol{W})$ satisfies local strong convexity and smoothness in $\mathbb{B}(\boldsymbol{W}^\star, r)$.

We will first show that the Hessian of the population risk is smooth enough around $\boldsymbol{W}^\star$ in the following lemma.

**Lemma 1.** *For sigmoid activations, assume $\|\boldsymbol{W}^\star\|_F \leq 1$, we have*

$$\|\nabla^2 f\left(\boldsymbol{W}\right) - \nabla^2 f\left(\boldsymbol{W}^\star\right)\| \leq \frac{C}{K^{\frac{1}{2}}} \cdot \|\boldsymbol{W} - \boldsymbol{W}^\star\|_{\mathrm{F}}, \tag{9}$$

*holds for some large enough constant C, when $\|\boldsymbol{W} - \boldsymbol{W}^\star\|_{\mathrm{F}} \leq 0.7$.*

The proof is given in Appendix D.2. Lemma 1 together with the fact that $\nabla^2 f(\boldsymbol{W}^\star)$ be lower and upper bounded, will allow us to bound $\nabla^2 f(\boldsymbol{W})$ in a neighborhood around ground truth, given below.

**Lemma 2** (Local Strong Convexity and Smoothness of Population Loss)**.** *For sigmoid activations, there exists some constant C, such that*

$$\frac{4}{K^2} \cdot \frac{\rho\left(\sigma_K\right)}{\kappa^2 \lambda} \cdot \boldsymbol{I} \preceq \nabla^2 f\left(\boldsymbol{W}\right) \preceq C \cdot \boldsymbol{I},$$

*holds for all $\boldsymbol{W} \in \mathbb{B}(\boldsymbol{W}^\star, r)$ with $r := \min\left\{\frac{C}{K^{\frac{3}{2}}} \cdot \frac{\rho(\sigma_K)}{\kappa^2\lambda}, 0.7\right\}$.*

The proof is given in Appendix D.3. The next step is to show the Hessian of the empirical loss function is close to the Hessian of the population loss function in a uniform sense, which can be summarized as following.

**Lemma 3.** *For sigmoid activations, there exists constant C such that as long as $n \geq C \cdot dK \log dK$, with probability at least $1 - d^{-10}$, the following holds*

$$\sup_{\boldsymbol{W} \in \mathbb{B}(\boldsymbol{W}^\star, r)} \|\nabla^2 f_n\left(\boldsymbol{W}\right) - \nabla^2 f\left(\boldsymbol{W}^\star\right)\| \leq C\sqrt{\frac{dK \log n}{n}},$$

*where $r := \min\left\{\frac{C}{K^{\frac{3}{2}}} \cdot \frac{\rho(\sigma_K)}{\kappa^2\lambda}, 0.7\right\}$.*

The proof can be found in Appendix D.4.

The final step is to combine Lemma 3 and Lemma 1 to obtain Theorem 1 as follows,

*Proof of Theorem 1.* By Lemma 3 and Lemma 2, we have with probability at least $1 - d^{-10}$,

$$\nabla^2 f_n(\boldsymbol{W}) \succeq \nabla^2 f\left(\boldsymbol{W}\right) - \left\|\nabla^2 f_n\left(\boldsymbol{W}\right) - \nabla^2 f(\boldsymbol{W})\right\| \cdot \boldsymbol{I}$$

$$\succeq \Omega\left(\frac{1}{K^2} \cdot \frac{\rho\left(\sigma_K\right)}{\kappa^2\lambda}\right) \cdot \boldsymbol{I} - \Omega\left(C \cdot \sqrt{\frac{dK \log n}{n}}\right) \cdot \boldsymbol{I}.$$

As long as the sample size $n$ is set such that

$$C \cdot \sqrt{\frac{dK \log n}{n}} \leq \frac{1}{K^2} \cdot \frac{\rho(\sigma_K)}{\kappa^2 \lambda},$$

i.e. $n \geq C \cdot dK^5 \log^2 d \cdot \left(\frac{\kappa^2 \lambda}{\rho(\sigma_K)}\right)^2$, we have

$$\nabla^2 f_n(\boldsymbol{W}) \succeq \Omega\left(\frac{1}{K^2} \cdot \frac{\rho(\sigma_K)}{\kappa^2 \lambda}\right) \cdot \boldsymbol{I}.$$

holds for all $\boldsymbol{W} \in \mathbb{B}(\boldsymbol{W}^\star, r)$. Similarly, we have

$$\nabla^2 f_n(\boldsymbol{W}) \preceq C \cdot \boldsymbol{I}$$

holds for all $\boldsymbol{W} \in \mathbb{B}(\boldsymbol{W}^\star, r)$. $\qquad \square$

## B  PROOF OF THEOREM 2

We have established that $f_n(\boldsymbol{W})$ is strongly convex in $\mathbb{B}(\boldsymbol{W}^\star, r)$ in Theorem 1, thus there exists at most one critical point in $\mathbb{B}(\boldsymbol{W}^\star, r)$. The proof of Theorem 2 follows the steps below:

1. We first show that the gradient $\nabla f_n(\boldsymbol{W})$ concentrates around $\nabla f(\boldsymbol{W})$ in $\mathbb{B}(\boldsymbol{W}^\star, r)$ (Lemma 4), and then invoke (Mei et al., 2016, Theorem 2) to guarantee there indeed exists a critical point $\widehat{\boldsymbol{W}}_n$ in $\mathbb{B}(\boldsymbol{W}^\star, r)$;

2. We next show $\widehat{\boldsymbol{W}}_n$ is close to $\boldsymbol{W}^\star$ and gradient descent converges linearly to $\widehat{\boldsymbol{W}}_n$ with a properly chosen step size.

The following lemma establishes that $\nabla f_n(\boldsymbol{W})$ uniformly concentrates around $\nabla f(\boldsymbol{W})$.

**Lemma 4.** *For sigmoid activation function, assume $\|\boldsymbol{W}^\star\|_F \leq 1$, there exists constant $C$ such that as long as $n \geq CdK \log(dK)$, with probability at least $1 - d^{-10}$, the following holds*

$$\sup_{\boldsymbol{W} \in \mathbb{B}(\boldsymbol{W}^\star, r)} \|\nabla f_n(\boldsymbol{W}) - \nabla f(\boldsymbol{W})\| \leq C\sqrt{\frac{d\sqrt{K} \log n}{n}},$$

*where $r := \min\left\{\frac{C}{K^{\frac{3}{2}}} \cdot \frac{\rho(\sigma_K)}{\kappa^2 \lambda}, 0.7\right\}$.*

Notice that for the population risk function, $f(\boldsymbol{W})$, $\boldsymbol{W}^\star$ is the unique critical point in $\mathbb{B}(\boldsymbol{W}^\star, r)$ due to local strong convexity. With Lemma 3 and Lemma 4, we can invoke (Mei et al., 2016, Theorem 2), which guarantees the following.

**Corollary 1.** *There exists one and only one critical point $\widehat{\boldsymbol{W}}_n \in \mathbb{B}(\boldsymbol{W}^*, r)$ that satisfies $\nabla f_n\left(\widehat{\boldsymbol{W}}_n\right) = \boldsymbol{0}$.*

We first show that $\widehat{\boldsymbol{W}}_n$ is close to $\boldsymbol{W}^\star$. By the intermediate value theorem, $\exists \boldsymbol{W}' \in \mathbb{B}(\boldsymbol{W}^\star, r)$ such that

$$
\begin{aligned}
f_n\left(\widehat{\boldsymbol{W}}_n\right) &= f_n(\boldsymbol{W}^\star) + \left\langle \nabla f_n(\boldsymbol{W}^\star), \text{vec}\left(\widehat{\boldsymbol{W}}_n - \boldsymbol{W}^\star\right)\right\rangle \\
&\quad + \frac{1}{2}\text{vec}\left(\widehat{\boldsymbol{W}}_n - \boldsymbol{W}^\star\right)^\top \nabla^2 f_n(\boldsymbol{W}') \text{vec}\left(\widehat{\boldsymbol{W}}_n - \boldsymbol{W}^\star\right) \\
&\leq f_n(\boldsymbol{W}^\star),
\end{aligned}
\tag{10}
$$

where the last inequality follows from the optimality of $\widehat{\boldsymbol{W}}_n$. By Theorem 1, we have

$$\frac{1}{2}\text{vec}\left(\widehat{\boldsymbol{W}}_n - \boldsymbol{W}^\star\right)^\top \nabla^2 f_n(\boldsymbol{W}') \text{vec}\left(\widehat{\boldsymbol{W}}_n - \boldsymbol{W}^\star\right) \geq \Omega\left(\frac{1}{K^2} \cdot \frac{\rho(\sigma_K)}{\kappa^2 \lambda}\right) \left\|\widehat{\boldsymbol{W}}_n - \boldsymbol{W}^\star\right\|_F^2.$$

$$\tag{11}$$

On the other hand, by the Cauchy-Schwarz inequality, we have

$$\left| \left\langle \nabla f_n \left( \boldsymbol{W}^\star \right), \text{vec} \left( \widehat{\boldsymbol{W}}_n - \boldsymbol{W}^\star \right) \right\rangle \right| \le \|\nabla f_n \left( \boldsymbol{W}^\star \right) \|_2 \|\widehat{\boldsymbol{W}}_n - \boldsymbol{W}^\star \|_\mathrm{F}$$

$$\le \Omega \left( \sqrt{\frac{dK^{1/2} \log n}{n}} \right) \|\widehat{\boldsymbol{W}}_n - \boldsymbol{W}^\star \|_\mathrm{F}, \qquad (12)$$

where the last line follows from Lemma 4. Plugging (11) and (12) into (10), we have

$$\|\widehat{\boldsymbol{W}}_n - \boldsymbol{W}^\star \|_\mathrm{F} \le \Omega \left( \frac{K^{\frac{9}{4}} \kappa^2 \lambda}{\rho \left( \sigma_K \right)} \sqrt{\frac{d \log n}{n}} \right). \qquad (13)$$

Now we have established there indeed exists a critical point in $\mathbb{B}(\boldsymbol{W}^\star, r)$. We can establish local linear convergence of gradient descent as below. Let $\boldsymbol{W}_t$ be the estimate at the $t$-th iteration. According to the update rule, we have

$$\left\| \boldsymbol{W}_{t+1} - \widehat{\boldsymbol{W}}_n \right\|_\mathrm{F}^2 = \left\| \boldsymbol{W}_t - \eta \nabla f_n \left( \boldsymbol{W}_t \right) - \widehat{\boldsymbol{W}}_n \right\|_\mathrm{F}^2$$

$$= \|\boldsymbol{W}_t - \widehat{\boldsymbol{W}}_n\|_\mathrm{F}^2 + \eta^2 \|\nabla f_n \left( \boldsymbol{W}_t \right) \|_\mathrm{F}^2 - 2\eta \left\langle \nabla f_n \left( \boldsymbol{W}_t \right), \text{vec} \left( \boldsymbol{W}_t - \widehat{\boldsymbol{W}}_n \right) \right\rangle. \qquad (14)$$

Moreover, by the fundamental theorem of calculus (Lang, 1993), $\nabla f_n \left( \boldsymbol{W}_t \right)$ can be written as

$$\nabla f_n \left( \boldsymbol{W}_t \right) = \nabla f_n \left( \boldsymbol{W}_t \right) - \nabla f_n \left( \widehat{\boldsymbol{W}}_n \right)$$

$$= \left( \int_0^1 \nabla^2 f_n \left( \widehat{\boldsymbol{W}}_n + \gamma \left( \boldsymbol{W}_t - \widehat{\boldsymbol{W}}_n \right) \right) d\gamma \right) \text{vec} \left( \boldsymbol{W}_t - \widehat{\boldsymbol{W}}_n \right),$$

where $\boldsymbol{W}(\gamma) = \widehat{\boldsymbol{W}}_n + \gamma \left( \boldsymbol{W}_t - \widehat{\boldsymbol{W}}_n \right)$ for $\gamma \in [0, 1]$. By Theorem 1, we have

$$H_\mathrm{min} \cdot \boldsymbol{I} \preceq \nabla^2 f_n \left( \boldsymbol{W}(\gamma) \right) \preceq H_\mathrm{max} \cdot \boldsymbol{I},$$

where $H_\mathrm{min} = \Omega \left( \frac{1}{K^2} \cdot \frac{\rho(\sigma_K)}{\kappa^2 \lambda} \right)$ and $H_\mathrm{max} = C$. Therefore, we have

$$\|\nabla f_n \left( \boldsymbol{W}_t \right) \|_\mathrm{F}^2 \le H_\mathrm{max}^2 \left\| \boldsymbol{W}_t - \widehat{\boldsymbol{W}}_n \right\|_\mathrm{F}^2.$$

Hence,

$$\|\boldsymbol{W}_{t+1} - \widehat{\boldsymbol{W}}_n\|_\mathrm{F}^2 \le \left( 1 - 2\eta H_\mathrm{min} + \eta^2 H_\mathrm{max}^2 \right) \|\boldsymbol{W}_t - \widehat{\boldsymbol{W}}_n\|_\mathrm{F}^2$$

$$\le \left( 1 - \frac{1}{2} \eta H_\mathrm{min} \right)^2 \|\boldsymbol{W}_t - \widehat{\boldsymbol{W}}_n\|_\mathrm{F}^2$$

as long as we set $\eta < \frac{H_\mathrm{min}}{H_\mathrm{max}^2} := \Omega \left( \frac{1}{K^2} \cdot \frac{\rho(\sigma_K)}{\kappa^2 \lambda} \right)$. In summary, gradient descent converges linearly to the local minimizer $\widehat{\boldsymbol{W}}_n$.

## C  PROOF OF THEOREM 3

The proof contains two parts. Part (a) proves that the estimation of the direction of $\boldsymbol{W}^\star$ is sufficiently accurate, which follows the arguments similar to those in (Zhong et al., 2017b) and is only briefly summarized below. Part (b) is different, where we do not require the homogeneous condition for the activation function, and instead, our proof is based on a mild condition in Assumption 2. We detail our proof in part (b).

We first define a tensor operation as follows. For a tensor $\boldsymbol{T} \in \mathbb{R}^{n_1 \times n_2 \times n_3}$ and three matrices $\boldsymbol{A} \in \mathbb{R}^{n_1 \times d_1}, \boldsymbol{B} \in \mathbb{R}^{n_2 \times d_2}, \boldsymbol{C} \in \mathbb{R}^{n_3 \times d_3}$, the $(i, j, k)$-th entry of the tensor $\boldsymbol{T} \left( \boldsymbol{A}, \boldsymbol{B}, \boldsymbol{C} \right)$ is given by

$$\sum_{i'}^{n_1} \sum_{j'}^{n_2} \sum_{k'}^{n_3} \boldsymbol{T}_{i',j',k'} \boldsymbol{A}_{i',i} \boldsymbol{B}_{j',j} \boldsymbol{C}_{k',k}. \qquad (15)$$

(a) In order to estimate the direction of each $\boldsymbol{w}_i$ for $i = 1, \ldots, K$, (Zhong et al., 2017b) shows that for the regression problem, if the sample size $n \geq d\text{poly}\,(K, \kappa, t, \log d)$, then

$$\|\overline{\boldsymbol{w}_i}^\star - s_i \boldsymbol{V}\widehat{\boldsymbol{u}}_i\| \leq \epsilon\text{poly}\,(\text{K}, \kappa) \tag{16}$$

holds with high probability. Such a result also holds for the classification problem with only slight difference in the proof as we describe as follows. The main idea of the proof is to bound the estimation error of $\boldsymbol{P}_2$ and $\boldsymbol{R}_3$ via Bernstein inequality. For the regression problem, Bernstein inequality was applied to terms associated with each neuron individually, and the bounds were then put together via triangle inequality in (Zhong et al., 2017b), whereas for the classification problem here, we apply Bernstein inequality to terms associated with all neurons all together. Another difference is that the label $y_i$ of the classification model is bounded by nature, whereas the output $y_i$ in the regression model needs to be upper bounded via homogeneously bounded conditions of the activation function. A reader can refer to (Zhong et al., 2017b) for the details of the proof for this part.

(b) In order to estimate $\|\boldsymbol{w}_i\|$ for $i = 1, \ldots, K$, we provide a different proof from (Zhong et al., 2017b), which does not require the homogeneous condition on the activation function, but assumes a more relaxed condition in Assumption 2.

We define a quantity $Q_1$ as follows:

$$Q_1 = \boldsymbol{M}_{l_1}(\boldsymbol{I}, \underbrace{\boldsymbol{\alpha}, \cdots, \boldsymbol{\alpha}}_{(l_1 - 1)}), \tag{17}$$

where $l_1$ is the first non-zero index such that $\boldsymbol{M}_{l_1} \neq 0$. For example, if $l_1 = 3$, then $Q_1$ takes the following form

$$Q_1 = \boldsymbol{M}_3\,(\boldsymbol{I}, \boldsymbol{\alpha}, \boldsymbol{\alpha}) = \frac{1}{K}\sum_{i=1}^K m_{3,i}(\|\boldsymbol{w}_i^\star\|)\left(\boldsymbol{\alpha}^\top \overline{\boldsymbol{w}}_i^\star\right)^2 \overline{\boldsymbol{w}}_i^\star, \tag{18}$$

where $\overline{\boldsymbol{w}} = \boldsymbol{w}/\|\boldsymbol{w}\|$ and by definition

$$m_{3,i}(\|\boldsymbol{w}_i^\star\|) = \mathbb{E}\left[\phi\left(\|\boldsymbol{w}_i^\star\| \cdot z\right)z^3\right] - 3\mathbb{E}\left[\phi\left(\|\boldsymbol{w}_i^\star\| \cdot z\right)z\right]. \tag{19}$$

Clearly, $Q_1$ has information of $\|\boldsymbol{w}_i^\star\|$, which can be estimated by solving the following optimization problem:

$$\beta^\star = \text{argmin}_{\beta \in \mathbb{R}^K}\left\|\frac{1}{K}\sum_{i=1}^K \beta_i s_i \overline{\boldsymbol{w}}_i^\star - Q_1\right\|, \tag{20}$$

where each entry of the solution takes the form

$$\beta_i^\star = s_i^3 m_{3,i}(\|\boldsymbol{w}_i^\star\|)\left(\boldsymbol{\alpha}^T s_i \overline{\boldsymbol{w}}_i^\star\right)^2. \tag{21}$$

In the initialization, we substitute $\widehat{Q}_1$ (estimated from training data) for $Q_1$, $\boldsymbol{V}\widehat{\boldsymbol{u}}_i$ (estimated in part (a)) for $s_i\overline{\boldsymbol{w}}_i^\star$ into (20), and obtain an estimate $\widehat{\beta}$ of $\beta^\star$. We then substitute $\widehat{\beta}$ for $\beta^\star$ and $\boldsymbol{V}\widehat{\boldsymbol{u}}_i$ for $s_i\overline{\boldsymbol{w}}_i^\star$ into (21) to obtain an estimate $\widehat{a}_i$ of $\|\boldsymbol{w}_i^\star\|$ via the following equation

$$\widehat{\beta}_i = s_i^3 m_{3,i}(\widehat{a}_i)\left(\boldsymbol{\alpha}^T \boldsymbol{V}\widehat{\boldsymbol{u}}_i\right)^2. \tag{22}$$

Furthermore, since $m_{l_1,i}(x)$ has fixed sign for $x > 0$ and for $l_1 \geq 1$, $s_i$ can be estimated correctly from the sign of $\widehat{\beta}_i$ for $i = 1, \ldots, K$.

For notational simplicity, let $\beta_{1,i}^\star := \frac{\beta_i^\star}{s_i^3(\boldsymbol{\alpha}^T s_i \overline{\boldsymbol{w}}_i^\star)^2}$ and $\widehat{\beta}_{1,i} := \frac{\widehat{\beta}_i}{s_i^3(\boldsymbol{\alpha}^T \boldsymbol{V}\widehat{\boldsymbol{u}}_i)^2}$, and then (21) and (22) become

$$\widehat{\beta}_{1,i} = m_{3,i}(\widehat{a}_i), \quad \beta_{1,i}^\star = m_{3,i}(\|\boldsymbol{w}_i^\star\|). \tag{23}$$

By Assumption 2 and (21), there exists a constant $\delta' > 0$ such that the inverse function $g(\cdot)$ of $m_{3,1}(\cdot)$ has upper-bounded derivative in the interval $(\beta_{1,i}^\star - \delta', \beta_{1,i}^\star + \delta')$, i.e., $|g'(x)| < \Gamma$ for a constant $\Gamma$. By employing the result in (Zhong et al., 2017b), if the sample size $n \geq d\text{poly}\,(K, \kappa, t, \log d)$, then $\widehat{Q}_1$ and $Q_1$, $\boldsymbol{V}\widehat{\boldsymbol{u}}_i$ and $s_i\overline{\boldsymbol{w}}_i^\star$ can be arbitrarily close so that $|\beta_{1,i}^\star - \widehat{\beta}_{1,i}| < \min\{\delta', \frac{r}{\sqrt{K}\Gamma}\}$.

Thus, by (23) and mean value theorem, we obtain

$$|\widehat{a}_i - \|\boldsymbol{w}_i^\star\|| = |g'(\xi)||\beta_{1,i}^\star - \widehat{\beta}_{1,i}| \tag{24}$$

where $\xi$ is between $\beta_{1,i}^\star$ and $\widehat{\beta}_{1,i}$, and hence $|g'(\xi)| < \Gamma$. Therefore, $|\widehat{a}_i - \|\boldsymbol{w}_i^\star\|| \leq \frac{r}{\sqrt{K}}$, which is the desired result.

# D  PROOF OF TECHNICAL LEMMAS

## D.1  PRELIMINARIES

We introduce some useful definitions and results that will be used in the proofs. The first one is the definition of norms of random variable, i.e.

**Definition 4** (Sub-gaussian and Sub-exponential norm). *The sub-gaussian norm of a random variable $X$, denotes as $\|X\|_{\psi_2}$, is defined as*

$$\|X\|_{\psi_2} = \sup_{p \geq 1} p^{-\frac{1}{2}} \left( \mathbb{E}\left[|X|^p\right]\right)^{\frac{1}{p}}, \tag{25}$$

*and the sub-exponential norm of $X$, denoted as $\|X\|_{\psi_1}$, is defined as*

$$\|X\|_{\psi_1} = \sup_{p \geq 1} p^{-1} \left( \mathbb{E}\left[|X|^p\right]\right)^{\frac{1}{p}}. \tag{26}$$

The definition is summarized from (Vershynin, 2012, Def 5.7,Def 5.13), and if $\|X\|_{\psi_2}$ is upper bounded, then $X$ is a sub-gaussian random variable and it satisfies

$$\mathbb{P}\left(|X| > t\right) \leq \exp\left(1 - ct^2/\|X\|_{\psi_2}^2\right) \text{ for all } t \geq 0. \tag{27}$$

Next we provide the calculations of the gradient and Hessian of $\mathbb{E}\left[\ell\left(\boldsymbol{W};\boldsymbol{x}\right)\right]$. Let's denote $p\left(\boldsymbol{W}\right) = \frac{1}{K}\sum_{i=1}^{K} \phi\left(\boldsymbol{w}_i^\top \boldsymbol{x}\right)$, and then

$$\mathbb{E}\left[\frac{\partial \ell\left(\boldsymbol{W};\boldsymbol{x}\right)}{\partial \boldsymbol{w}_j}\right] = \mathbb{E}\left[-\frac{1}{K}\left(\frac{\frac{1}{K}\sum_{i=1}^{K}\phi\left(\boldsymbol{w}_i^{\star\top}\boldsymbol{x}\right) - \frac{1}{K}\sum_{i=1}^{K}\phi\left(\boldsymbol{w}_i^\top\boldsymbol{x}\right)}{\left(\frac{1}{K}\sum_{i=1}^{K}\phi\left(\boldsymbol{w}_i^\top\boldsymbol{x}\right)\right)\left(1 - \frac{1}{K}\sum_{i=1}^{K}\phi\left(\boldsymbol{w}_i^\top\boldsymbol{x}\right)\right)}\cdot\phi'\left(\boldsymbol{w}_j^\top\boldsymbol{x}\right)\right)\boldsymbol{x}\right],$$

$$= \mathbb{E}\left[-\frac{1}{K}\phi'\left(\boldsymbol{w}_j^\top\boldsymbol{x}\right)\cdot\frac{p\left(\boldsymbol{W}^\star\right) - p\left(\boldsymbol{W}\right)}{p\left(\boldsymbol{W}\right)\left(1 - p\left(\boldsymbol{W}\right)\right)}\cdot\boldsymbol{x}\right] \tag{28}$$

$$\mathbb{E}\left[\frac{\nabla^2 \ell\left(\boldsymbol{W};\boldsymbol{x}\right)}{\partial \boldsymbol{w}_j \partial \boldsymbol{w}_l}\right] = \mathbb{E}\left[\frac{\xi_{j,l}\left(\boldsymbol{W}\right)}{\left(p\left(\boldsymbol{W}\right)\left(1 - p\left(\boldsymbol{W}\right)\right)\right)^2}\cdot\boldsymbol{x}\boldsymbol{x}^\top\right] \tag{29}$$

where if $j \neq l$,

$$\xi_{j,l}\left(\boldsymbol{W}\right) = \frac{1}{K^2}\phi'\left(\boldsymbol{w}_j^\top\boldsymbol{x}\right)\phi'\left(\boldsymbol{w}_l^\top\boldsymbol{x}\right)\cdot\left(p\left(\boldsymbol{W}\right)^2 + p\left(\boldsymbol{W}^\star\right) - 2p\left(\boldsymbol{W}^\star\right)p\left(\boldsymbol{W}\right)\right),$$

and if $j = l$,

$$\xi_{j,j}\left(\boldsymbol{W}\right) = \frac{1}{K^2}\phi'\left(\boldsymbol{w}_j^\top\boldsymbol{x}\right)^2\cdot\left(p\left(\boldsymbol{W}\right)^2 + p\left(\boldsymbol{W}^\star\right) - 2p\left(\boldsymbol{W}^\star\right)p\left(\boldsymbol{W}\right)\right)$$

$$- \frac{1}{K}\phi''\left(\boldsymbol{w}_j^\top\boldsymbol{x}\right)\left(p\left(\boldsymbol{W}^\star\right) - p\left(\boldsymbol{W}\right)\right)\left(p\left(\boldsymbol{W}\right)\left(1 - p\left(\boldsymbol{W}\right)\right)\right).$$

## D.2  PROOF OF LEMMA 1

*Proof.* Let $\boldsymbol{\Delta} = \nabla^2 f(\boldsymbol{W}) - \nabla^2 f(\boldsymbol{W}^\star)$. For each $(j,l) \in [K] \times [K]$, let $\boldsymbol{\Delta}_{j,l} \in \mathbb{R}^{d \times d}$ denote the $(j,l)$-th block of $\boldsymbol{\Delta}$. Let $\boldsymbol{a} = [\boldsymbol{a}_1^\top, \cdots, \boldsymbol{a}_K^\top]^\top \in \mathbb{R}^{dK}$. Since by definition,

$$\|\nabla^2 f(\boldsymbol{W}) - \nabla^2 f(\boldsymbol{W}^\star)\| = \max_{\|\boldsymbol{a}\|=1} \boldsymbol{a}^\top(\nabla^2 f(\boldsymbol{W}) - \nabla^2 f(\boldsymbol{W}^\star))\boldsymbol{a}$$

$$= \max_{\|\boldsymbol{a}\|=1} \sum_{j=1}^{K}\sum_{l=1}^{K} \boldsymbol{a}_j^\top \boldsymbol{\Delta}_{j,l} \boldsymbol{a}_l. \tag{30}$$

Next we will evaluate $\boldsymbol{\Delta}_{j,l}$. From (29) we can write the hessian block more concisely as

$$\frac{\partial^2 f\left(\boldsymbol{W}\right)}{\partial \boldsymbol{w}_j \partial \boldsymbol{w}_l} = \mathbb{E}\left[g_{j,l}\left(\boldsymbol{W}\right) \cdot \boldsymbol{x}\boldsymbol{x}^\top\right], \tag{31}$$

where $g_{j,l}\left(\boldsymbol{W}\right) = \frac{\xi_{j,l}(\boldsymbol{W})}{(p(\boldsymbol{W})(1-p(\boldsymbol{W})))^2} \in \mathbb{R}$, and then by the mean value theorem, we can write $g_{j,l}\left(\boldsymbol{W}\right)$ as

$$g_{j,l}\left(\boldsymbol{W}\right) = g_{j,l}\left(\boldsymbol{W}^\star\right) + \sum_{k=1}^{K}\left\langle \frac{\partial g_{j,l}\left(\widetilde{\boldsymbol{W}}\right)}{\partial \widetilde{\boldsymbol{w}}_k}, \boldsymbol{w}_k - \boldsymbol{w}_k^\star \right\rangle \tag{32}$$

where $\widetilde{\boldsymbol{W}} = \eta \cdot \boldsymbol{W} + (1-\eta)\,\boldsymbol{W}^\star$ for some $\eta \in (0,1)$. Thus we can calculate $\boldsymbol{\Delta}_{j,l}$ as

$$\begin{aligned}
\boldsymbol{\Delta}_{j,l} &= \frac{\partial^2 f\left(\boldsymbol{W}\right)}{\partial \boldsymbol{w}_j \partial \boldsymbol{w}_l} - \frac{\partial^2 f\left(\boldsymbol{W}^\star\right)}{\partial \boldsymbol{w}_j^\star \partial \boldsymbol{w}_l^\star} \\
&= \mathbb{E}\left[g_{j,l}\left(\boldsymbol{W}\right) \cdot \boldsymbol{x}\boldsymbol{x}^\top\right] - \mathbb{E}\left[g_{j,l}\left(\boldsymbol{W}^\star\right) \cdot \boldsymbol{x}\boldsymbol{x}^\top\right] \\
&= \mathbb{E}\left[\left(\sum_{k=1}^{K}\left\langle \frac{\partial g_{j,l}\left(\widetilde{\boldsymbol{W}}\right)}{\partial \widetilde{\boldsymbol{w}}_k}, \boldsymbol{w}_k - \boldsymbol{w}_k^\star \right\rangle\right) \cdot \boldsymbol{x}\boldsymbol{x}^\top\right],
\end{aligned} \tag{33}$$

and plug it back to (30) we can obtain

$$\begin{aligned}
&\left\|\nabla^2 f(\boldsymbol{W}) - \nabla^2 f(\boldsymbol{W}^\star)\right\| \\
&= \max_{\|\boldsymbol{a}\|=1} \sum_{j=1}^{K}\sum_{l=1}^{K} \boldsymbol{a}_j^\top \boldsymbol{\Delta}_{j,l} \boldsymbol{a}_l \\
&= \max_{\|\boldsymbol{a}\|=1} \sum_{j=1}^{K}\sum_{l=1}^{K} \mathbb{E}\left[\left(\sum_{k=1}^{K}\left\langle \frac{\partial g_{j,l}\left(\widetilde{\boldsymbol{W}}\right)}{\partial \widetilde{\boldsymbol{w}}_k}, \boldsymbol{w}_k - \boldsymbol{w}_k^\star \right\rangle\right) \cdot \left(\boldsymbol{a}_j^\top \boldsymbol{x}\right)\left(\boldsymbol{a}_l^\top \boldsymbol{x}\right)\right] \\
&= \max_{\|\boldsymbol{a}\|=1} \sum_{j=1}^{K}\sum_{l=1}^{K} \mathbb{E}\left[\left(\sum_{k=1}^{K} T_{j,l,k}\left\langle \boldsymbol{x}, \boldsymbol{w}_k - \boldsymbol{w}_k^\star\right\rangle\right) \cdot \left(\boldsymbol{a}_j^\top \boldsymbol{x}\right)\left(\boldsymbol{a}_l^\top \boldsymbol{x}\right)\right] \\
&\leq \max_{\|\boldsymbol{a}\|=1} \sum_{j=1}^{K}\sum_{l=1}^{K} \sqrt{\mathbb{E}\left[\sum_{k=1}^{K} T_{j,l,k}^2\right]} \cdot \sqrt{\mathbb{E}\left[\sum_{k=1}^{K}\left(\langle \boldsymbol{x}, \boldsymbol{w}_k - \boldsymbol{w}_k^\star\rangle\right)^2 \left(\boldsymbol{a}_j^\top \boldsymbol{x}\right)^2 \left(\boldsymbol{a}_l^\top \boldsymbol{x}\right)^2\right]} \\
&\leq \max_{\|\boldsymbol{a}\|=1} \sum_{j=1}^{K}\sum_{l=1}^{K} \sqrt{\sum_{k=1}^{K}\mathbb{E}\left[T_{j,l,k}^2\right]} \cdot \sqrt{\sum_{k=1}^{K}\|\boldsymbol{w}_k - \boldsymbol{w}_k^\star\|_2^2 \cdot \|\boldsymbol{a}_j\|_2^2 \cdot \|\boldsymbol{a}_l\|_2^2},
\end{aligned} \tag{34}$$

for the third equality we have used the fact that $\frac{\partial g_{j,l}\left(\widetilde{\boldsymbol{W}}\right)}{\partial \widetilde{\boldsymbol{w}}_k}$ can be written as $T_{j,l,k} \cdot \boldsymbol{x}$, where $T_{j,l,k} \in \mathbb{R}$, since the variable of $g_{j,l}\left(\widetilde{\boldsymbol{W}}\right)$ is in the form of $\boldsymbol{w}_i^\top \boldsymbol{x}$. and for the last two inequalities, we have used Cauchy-Schwarz inequality. Our next goal is to upper bound $\mathbb{E}\left[T_{j,l,k}^2\right]$. Further since

$$\frac{\partial g_{j,l}\left(\boldsymbol{W}\right)}{\partial \boldsymbol{w}_k} = \frac{1}{K^2} \cdot \frac{\partial \frac{\phi'\left(\boldsymbol{w}_j^\top \boldsymbol{x}\right)\phi'\left(\boldsymbol{w}_l^\top \boldsymbol{x}\right)\cdot\left(p(\boldsymbol{W})^2 + p(\boldsymbol{W}^\star) - 2p(\boldsymbol{W}^\star)p(\boldsymbol{W})\right)}{(p(\boldsymbol{W})(1-p(\boldsymbol{W})))^2}}{\partial \boldsymbol{w}_k},$$

which aligns with $\boldsymbol{x}$ and the scalar coefficient is upper bounded by $\frac{1}{K^2} \cdot \frac{C}{\left(p(\widetilde{\boldsymbol{W}})(1-p(\widetilde{\boldsymbol{W}}))\right)^3}$, since $\phi\left(\cdot\right), \phi'\left(\cdot\right), \phi''\left(\cdot\right)$ are all upper bounded, thus we leave only the denominator. And then

$$\mathbb{E}\left[T_{j,l,k}^2\right] \leq \frac{C}{K^4} \cdot \mathbb{E}\left[\frac{1}{\left(p\left(\widetilde{\boldsymbol{W}}\right)\left(1 - p\left(\widetilde{\boldsymbol{W}}\right)\right)\right)^6}\right] \leq \frac{C}{K^4} \cdot e^{\|\widetilde{\boldsymbol{W}}\|_{\mathrm{F}}^2}, \tag{35}$$

holds for some constant $C$, where the second inequality follows from Lemma 5.

**Lemma 5.** *Let $x \sim \mathcal{N}(\mathbf{0}, \mathbf{I})$, $t = \max\{\|w_1\|_2, \cdots \|w_K\|_2\}$ and $z \in \mathbb{Z}$ such that $z \geq 1$, for the sigmoid activation function $\phi(x) = \frac{1}{1+e^{-x}}$, the following*

$$\mathbb{E}\left[\left(\frac{1}{\frac{1}{K}\sum_{i=1}^{K}\phi\left(w_i^\top x\right)\left(1 - \frac{1}{K}\sum_{i=1}^{K}\phi\left(w_i^\top x\right)\right)}\right)^z\right] \leq C \cdot e^{t^2}, \tag{36}$$

*holds for a large enough constant $C$ which depends on the constant $z$.*

Plugging (35) into (34), we can obtain

$$\|\nabla^2 f(\boldsymbol{W}) - \nabla^2 f(\boldsymbol{W}^\star)\| \leq \frac{C}{K^{\frac{3}{2}}} e^{\|\widetilde{\boldsymbol{W}}\|_{\mathrm{F}}^2} \cdot \|\boldsymbol{W} - \boldsymbol{W}^\star\|_{\mathrm{F}} \cdot \max_{\|\boldsymbol{a}\|=1} \sum_{j=1}^{K}\sum_{l=1}^{K}\|\boldsymbol{a}_j\|_2\|\boldsymbol{a}_l\|_2$$

$$\leq \frac{C}{K^{\frac{1}{2}}} e^{\|\widetilde{\boldsymbol{W}}\|_{\mathrm{F}}^2} \cdot \|\boldsymbol{W} - \boldsymbol{W}^\star\|_{\mathrm{F}}, \tag{37}$$

Further since $e^{\|\widetilde{\boldsymbol{W}}\|_{\mathrm{F}}^2} \leq C \cdot (1 + \|\boldsymbol{W} - \boldsymbol{W}^\star\|_{\mathrm{F}})$ when $\|\boldsymbol{W} - \boldsymbol{W}^\star\|_{\mathrm{F}} \leq 0.7$, where we have used the assumption that $\|\boldsymbol{W}^\star\|_F \leq 1$ thus we can conclude that if $\|\boldsymbol{W} - \boldsymbol{W}^\star\|_{\mathrm{F}} \leq 0.7$, then

$$\|\nabla^2 f(\boldsymbol{W}) - \nabla^2 f(\boldsymbol{W}^\star)\| \leq \frac{C}{K^{\frac{1}{2}}}\|\boldsymbol{W} - \boldsymbol{W}^\star\|_{\mathrm{F}} \tag{38}$$

holds for some constant $C$. $\qquad\square$

### D.3 PROOF OF LEMMA 2

*Proof.* We will first present upper and lower bounds of the Hessian of the population risk at ground truth, i.e. $\nabla^2 f(\boldsymbol{W}^\star)$, and then apply Lemma 1 to obtain a uniform bound in the neighborhood of $\boldsymbol{W}^\star$. As a reminder,

$$\frac{\partial^2 f(\boldsymbol{W}^\star)}{\partial w_j^2} = \mathbb{E}\left[\frac{1}{K^2} \cdot \left(\frac{\phi'\left(w_j^{\star\top}x\right)^2}{\left(\frac{1}{K}\sum_{i=1}^{K}\phi\left(w_i^{\star\top}x\right)\right)\left(1 - \frac{1}{K}\sum_{i=1}^{K}\phi\left(w_i^{\star\top}x\right)\right)}\right)xx^\top\right] \tag{39}$$

$$\frac{\partial^2 f(\boldsymbol{W}^\star)}{\partial w_j \partial w_l} = \mathbb{E}\left[\frac{1}{K^2} \cdot \left(\frac{\phi'\left(w_j^{\star\top}x\right)\phi'\left(w_l^{\star\top}x\right)}{\left(\frac{1}{K}\sum_{i=1}^{K}\phi\left(w_i^{\star\top}x\right)\right)\left(1 - \frac{1}{K}\sum_{i=1}^{K}\phi\left(w_i^{\star\top}x\right)\right)}\right)xx^\top\right], \tag{40}$$

and let $\boldsymbol{a} = [\boldsymbol{a}_1^\top, \cdots, \boldsymbol{a}_K^\top]^\top \in \mathbb{R}^{dK}$, we can write

$$\nabla^2 f(\boldsymbol{W}^\star) \succeq \left(\min_{\|\boldsymbol{a}\|_2=1} \boldsymbol{a}^\top \nabla^2 f(\boldsymbol{W}^\star)\boldsymbol{a}\right) \cdot \boldsymbol{I}$$

$$= \min_{\|\boldsymbol{a}\|_2=1} \frac{1}{K^2}\mathbb{E}\left[\frac{\left(\sum_{i=1}^{K}\phi'\left(w_i^{\star\top}x\right)\left(\boldsymbol{a}_i^\top x\right)\right)^2}{\left(\frac{1}{K}\sum_{i=1}^{K}\phi\left(w_i^{\star\top}x\right)\right)\left(1 - \frac{1}{K}\sum_{i=1}^{K}\phi\left(w_i^{\star\top}x\right)\right)}\right]$$

$$\succeq \min_{\|\boldsymbol{a}\|_2=1} \frac{4}{K^2}\mathbb{E}\left[\left(\sum_{i=1}^{K}\phi'\left(w_i^{\star\top}x\right)\left(\boldsymbol{a}_i^\top x\right)\right)^2\right]$$

$$\succeq \frac{4}{K^2} \cdot \frac{\rho(\sigma_K)}{\kappa^2\lambda} \cdot \boldsymbol{I}, \tag{41}$$

the second inequality holds due to the fact that $\left(\frac{1}{K}\sum_{i=1}^{K}\phi\left(w_i^{\star\top}x\right)\right)\left(1 - \frac{1}{K}\sum_{i=1}^{K}\phi\left(w_i^{\star\top}x\right)\right) \leq \frac{1}{4}$, and the last inequality follows from (Zhong et al., 2017b, Lemmas D.4 and D.6).

Further more, we can uppder bound $\nabla^2 f(\boldsymbol{W}^\star)$ as

$$
\nabla^2 f(\boldsymbol{W}^\star)
$$
$$
\preceq \left( \max_{\|\boldsymbol{a}\|_2 = 1} \boldsymbol{a}^\top \nabla^2 f(\boldsymbol{W}^\star) \boldsymbol{a} \right) \cdot \boldsymbol{I}
$$
$$
= \max_{\|\boldsymbol{a}\|_2=1} \frac{1}{K^2} \mathbb{E} \left[ \frac{\left( \sum_{i=1}^K \phi'\left(\boldsymbol{w}_i^{\star\top}\boldsymbol{x}\right)\left(\boldsymbol{a}_i^\top \boldsymbol{x}\right) \right)^2}{\frac{1}{K^2} \sum_{i=1}^K \sum_{j=1}^K \phi\left(\boldsymbol{w}_i^{\star\top}\boldsymbol{x}\right)\left(1 - \phi\left(\boldsymbol{w}_j^{\star\top}\boldsymbol{x}\right)\right)} \right] \cdot \boldsymbol{I}
$$
$$
\preceq \max_{\|\boldsymbol{a}\|_2=1} \frac{1}{K^2} \mathbb{E} \left[ \frac{\left( \sum_{i=1}^K \phi'\left(\boldsymbol{w}_i^{\star\top}\boldsymbol{x}\right)^2 \right) \cdot \left( \sum_{i=1}^K \left(\boldsymbol{a}_i^\top \boldsymbol{x}\right)^2 \right)}{\frac{1}{K^2} \sum_{i=1}^K \sum_{j=1}^K \phi\left(\boldsymbol{w}_i^{\star\top}\boldsymbol{x}\right)\left(1 - \phi\left(\boldsymbol{w}_j^{\star\top}\boldsymbol{x}\right)\right)} \right] \cdot \boldsymbol{I} \quad \text{by Cauchy-Schwarz inequality}
$$
$$
\preceq \max_{\|\boldsymbol{a}\|_2=1} \frac{1}{K^2} \mathbb{E} \left[ \frac{\frac{C}{4}\left( \sum_{i=1}^K \phi'\left(\boldsymbol{w}_i^{\star\top}\boldsymbol{x}\right) \right) \cdot \left( \sum_{i=1}^K \left(\boldsymbol{a}_i^\top \boldsymbol{x}\right)^2 \right)}{\frac{1}{K^2} \sum_{i=1}^K \phi\left(\boldsymbol{w}_i^{\star\top}\boldsymbol{x}\right)\left(1 - \phi\left(\boldsymbol{w}_i^{\star\top}\boldsymbol{x}\right)\right)} \right] \cdot \boldsymbol{I}
$$
$$
\preceq \max_{\|\boldsymbol{a}\|_2=1} \frac{1}{K^2} \mathbb{E} \left[ \frac{CK^2}{4} \sum_{i=1}^K \left(\boldsymbol{a}_i^\top \boldsymbol{x}\right)^2 \right] \cdot \boldsymbol{I}
$$
$$
= C \cdot \boldsymbol{I}, \tag{42}
$$

where for the third and fourth inequality we have used the fact that $\phi\left(\boldsymbol{w}_i^{\star\top}\boldsymbol{x}\right)\left(1 - \phi\left(\boldsymbol{w}_i^{\star\top}\boldsymbol{x}\right)\right) \leq \frac{1}{4}$ and

$$
\sum_{i=1}^K \sum_{j=1}^K \phi\left(\boldsymbol{w}_i^{\star\top}\boldsymbol{x}\right)\left(1 - \phi\left(\boldsymbol{w}_j^{\star\top}\boldsymbol{x}\right)\right) \geq \sum_{i=1}^K \phi\left(\boldsymbol{w}_i^{\star\top}\boldsymbol{x}\right)\left(1 - \phi\left(\boldsymbol{w}_i^{\star\top}\boldsymbol{x}\right)\right) = \sum_{i=1}^K \phi'\left(\boldsymbol{w}_i^{\star\top}\boldsymbol{x}\right).
$$

Thus together with the lower bound (41) we can conclude that

$$
\frac{4}{K^2} \cdot \frac{\rho(\sigma_K)}{\kappa^2 \lambda} \cdot \boldsymbol{I} \preceq \nabla^2 f(\boldsymbol{W}^\star) \preceq C \cdot \boldsymbol{I}, \tag{43}
$$

From Lemma 1, we have

$$
\|\nabla^2 f(\boldsymbol{W}) - \nabla^2 f(\boldsymbol{W}^\star)\| \lesssim \frac{C}{K^{\frac{1}{2}}} \|\boldsymbol{W} - \boldsymbol{W}^\star\|_F, \tag{44}
$$

therefore, when $\|\boldsymbol{W}^\star - \boldsymbol{W}\|_F \leq 0.7$ and

$$
\frac{C}{K^{\frac{1}{2}}} \cdot \|\boldsymbol{W} - \boldsymbol{W}^\star\|_F \leq \frac{4}{K^2} \cdot \frac{\rho(\sigma_K)}{\kappa^2 \lambda},
$$

i.e., when $\|\boldsymbol{W} - \boldsymbol{W}^\star\|_F \leq \min\left\{ \frac{C}{K^{\frac{3}{2}}} \cdot \frac{\rho(\sigma_K)}{\kappa^2 \lambda}, 0.7 \right\}$ for some constant $C$, we have

$$
\sigma_{\min}\left(\nabla^2 f(\boldsymbol{W})\right) \geq \sigma_{\min}\left(\nabla^2 f(\boldsymbol{W}^\star)\right) - \|\nabla^2 f(\boldsymbol{W}) - \nabla^2 f(\boldsymbol{W}^\star)\| \tag{45}
$$
$$
\gtrsim \frac{4}{K^2} \cdot \frac{\rho(\sigma_K)}{\kappa^2 \lambda} - \frac{C}{K^{\frac{1}{2}}} \|\boldsymbol{W} - \boldsymbol{W}^\star\|_F \gtrsim \frac{4}{K^2} \cdot \frac{\rho(\sigma_K)}{\kappa^2 \lambda}. \tag{46}
$$

Moreover, within the same neighborhood, by the triangle inequality we have

$$
\|\nabla^2 f(\boldsymbol{W})\| \leq \|\nabla^2 f(\boldsymbol{W}) - \nabla^2 f(\boldsymbol{W}^\star)\| + \|\nabla^2 f(\boldsymbol{W}^\star)\| \lesssim C. \tag{47}
$$

$\square$

## D.4 PROOF OF LEMMA 3

*Proof.* We adapt the analysis in (Mei et al., 2016) to our setting. Let $N_\epsilon$ be the $\epsilon$-covering number of the Euclidean ball $\mathbb{B}(\boldsymbol{W}^\star, r)$. It is known that $\log N_\epsilon \leq dK \log(3r/\epsilon)$ (Vershynin, 2010). Let $\mathcal{W}_\epsilon = \{\boldsymbol{W}_1, \cdots, \boldsymbol{W}_{N_\epsilon}\}$ be the $\epsilon$-cover set with $N_\epsilon$ elements. For any $\boldsymbol{W} \in \mathbb{B}(\boldsymbol{W}^\star, r)$, let $j(\boldsymbol{W}) = \operatorname{argmin}_{j \in [N_\epsilon]} \|\boldsymbol{W} - \boldsymbol{W}_{j(\boldsymbol{W})}\|_F \leq \epsilon$ for all $\boldsymbol{W} \in \mathbb{B}(\boldsymbol{W}^\star, r)$.

For any $\boldsymbol{W} \in \mathbb{B}\left(\boldsymbol{W}^\star, r\right)$, we have

$$
\begin{aligned}
\left\|\nabla^2 f_n\left(\boldsymbol{W}\right) - \nabla^2 f(\boldsymbol{W})\right\| \leq & \frac{1}{n}\left\|\sum_{i=1}^n\left[\nabla^2 \ell\left(\boldsymbol{W}; \boldsymbol{x}_i\right) - \nabla^2 \ell\left(\boldsymbol{W}_{j(\boldsymbol{W})}; \boldsymbol{x}_i\right)\right]\right\| \\
& + \left\|\frac{1}{n}\sum_{i=1}^n \nabla^2 \ell\left(\boldsymbol{W}_{j(\boldsymbol{W})}; \boldsymbol{x}_i\right) - \mathbb{E}\left[\nabla^2 \ell\left(\boldsymbol{W}_{j(\boldsymbol{w})}; \boldsymbol{x}\right)\right]\right\| \\
& + \left\|\mathbb{E}\left[\nabla^2 \ell\left(\boldsymbol{W}_{j(\boldsymbol{W})}; \boldsymbol{x}\right)\right] - \mathbb{E}\left[\nabla^2 \ell\left(\boldsymbol{W}; \boldsymbol{x}\right)\right]\right\|.
\end{aligned}
$$

Hence, we have

$$
\mathbb{P}\left(\sup_{\boldsymbol{W} \in \mathbb{B}(\boldsymbol{W}^\star, r)}\left\|\nabla^2 f_n\left(\boldsymbol{W}\right) - \nabla^2 f(\boldsymbol{W})\right\| \geq t\right) \leq \mathbb{P}\left(A_t\right) + \mathbb{P}\left(B_t\right) + \mathbb{P}\left(C_t\right), \qquad (48)
$$

where the events $A_t$, $B_t$ and $C_t$ are defined as

$$
A_t = \left\{\sup_{\boldsymbol{W} \in \mathbb{B}(\boldsymbol{W}^\star, r)} \frac{1}{n}\left\|\sum_{i=1}^n\left[\nabla^2 \ell\left(\boldsymbol{W}; \boldsymbol{x}_i\right) - \nabla^2 \ell\left(\boldsymbol{W}_{j(\boldsymbol{W})}; \boldsymbol{x}_i\right)\right]\right\| \geq \frac{t}{3}\right\}, \qquad (49)
$$

$$
B_t = \left\{\sup_{\boldsymbol{W} \in \mathcal{W}_\epsilon}\left\|\frac{1}{n}\sum_{i=1}^n \nabla^2 \ell\left(\boldsymbol{W}; \boldsymbol{x}_i\right) - \mathbb{E}\left[\nabla^2 \ell\left(\boldsymbol{W}; \boldsymbol{x}\right)\right]\right\| \geq \frac{t}{3}\right\}, \qquad (50)
$$

$$
C_t = \left\{\sup_{\boldsymbol{W} \in \mathbb{B}(\boldsymbol{W}^\star, r)}\left\|\mathbb{E}\left[\nabla^2 \ell\left(\boldsymbol{W}_{j(\boldsymbol{W})}; \boldsymbol{x}\right)\right] - \mathbb{E}\left[\nabla^2 \ell\left(\boldsymbol{W}; \boldsymbol{x}\right)\right]\right\| \geq \frac{t}{3}\right\}. \qquad (51)
$$

In the sequel, we will bound the terms $\mathbb{P}\left(A_t\right)$, $\mathbb{P}\left(B_t\right)$, and $\mathbb{P}\left(C_t\right)$, separately.

1. **Upper bound $\mathbb{P}\left(B_t\right)$.** Before continuing, let us state a simple technical lemma that is useful for our proof, whose proof can be found in (Mei et al., 2016).

   **Lemma 6.** *Let $\boldsymbol{M} \in \mathbb{R}^{d \times d}$ be a symmetric $d \times d$ matrix and $V_\epsilon$ be an $\epsilon$-cover of unit-Euclidean-norm ball $\mathbb{B}\left(\mathbf{0}, 1\right)$, then*

   $$
   \|\boldsymbol{M}\| \leq \frac{1}{1 - 2\epsilon}\sup_{\boldsymbol{v} \in V_\epsilon}\left|\left\langle \boldsymbol{v}, \boldsymbol{M}\boldsymbol{v}\right\rangle\right|. \qquad (52)
   $$

   Let $V_{\frac{1}{4}}$ be a $\left(\frac{1}{4}\right)$-cover of the ball $\mathbb{B}(\mathbf{0}, 1) = \{\boldsymbol{W} \in \mathbb{R}^{d \times K} : \|\boldsymbol{W}\|_{\mathrm{F}} = 1\}$, where $\log|V_{\frac{1}{4}}| \leq dK \log 12$. From Lemma 6, we know that

   $$
   \left\|\frac{1}{n}\sum_{i=1}^n \nabla^2 \ell\left(\boldsymbol{W}; \boldsymbol{x}_i\right) - \mathbb{E}\left[\nabla^2 \ell\left(\boldsymbol{W}; \boldsymbol{x}\right)\right]\right\| \leq 2\sup_{\boldsymbol{v} \in V_{\frac{1}{4}}}\left|\left\langle \boldsymbol{v}, \left(\frac{1}{n}\sum_{i=1}^n \nabla^2 \ell\left(\boldsymbol{W}; \boldsymbol{x}_i\right) - \mathbb{E}\left[\nabla^2 \ell\left(\boldsymbol{W}; \boldsymbol{x}\right)\right]\right)\boldsymbol{v}\right\rangle\right|. \qquad (53)
   $$

   Taking the union bound over $\mathcal{W}_\epsilon$ and $V_{\frac{1}{4}}$ yields

   $$
   \begin{aligned}
   \mathbb{P}\left(B_t\right) \leq & \mathbb{P}\left(\sup_{\boldsymbol{W} \in \mathcal{W}_\epsilon, \boldsymbol{v} \in V_{\frac{1}{4}}}\left|\frac{1}{n}\sum_{i=1}^n\left\langle \boldsymbol{v}, \left(\nabla^2 \ell\left(\boldsymbol{W}; \boldsymbol{x}_i\right) - \mathbb{E}\left[\nabla^2 \ell\left(\boldsymbol{W}; \boldsymbol{x}\right)\right]\right)\boldsymbol{v}\right\rangle\right| \geq \frac{t}{6}\right) \\
   \leq & e^{dK\left(\log \frac{3r}{\epsilon} + \log 12\right)}\sup_{\boldsymbol{W} \in \mathcal{W}_\epsilon, \boldsymbol{v} \in V_{\frac{1}{4}}} \mathbb{P}\left(\left|\frac{1}{n}\sum_{i=1}^n\left\langle \boldsymbol{v}, \left(\nabla^2 \ell\left(\boldsymbol{W}; \boldsymbol{x}_i\right) - \mathbb{E}\left[\nabla^2 \ell\left(\boldsymbol{W}; \boldsymbol{x}\right)\right]\right)\boldsymbol{v}\right\rangle\right| \geq \frac{t}{6}\right).
   \end{aligned} \qquad (54)
   $$

   Let $G_i = \left\langle \boldsymbol{v}, \left(\nabla^2 \ell\left(\boldsymbol{W}; \boldsymbol{x}_i\right) - \mathbb{E}\left[\nabla^2 \ell\left(\boldsymbol{W}; \boldsymbol{x}\right)\right]\right)\boldsymbol{v}\right\rangle$ where $\mathbb{E}[G_i] = 0$. Let $\boldsymbol{a} = \left[\boldsymbol{a}_1^\top, \cdots, \boldsymbol{a}_K^\top\right] \in \mathbb{R}^{dK}$. Then we can show that $\|G_i\|_{\psi_1}$ is upper bounded, which we summariz as follows.

   **Lemma 7.** *There exists some constant $C$ such that*

   $$
   \|G_i\|_{\psi_1} \leq C :\equiv \tau^2.
   $$

Applying the Bernstein inequality for sub-exponential random variables (Mei et al., 2016, Theorem 9) to (54), we have for fixed $\boldsymbol{W} \in \mathcal{W}_\epsilon, \boldsymbol{v} \in V_{\frac{1}{4}}$,

$$\mathbb{P}\left(\left|\frac{1}{n}\sum_{i=1}^{n}\left\langle\boldsymbol{v},\left(\nabla^2\ell\left(\boldsymbol{W};\boldsymbol{x}_i\right)-\mathbb{E}\left[\nabla^2\ell\left(\boldsymbol{W};\boldsymbol{x}\right)\right]\right)\boldsymbol{v}\right\rangle\right|\geq\frac{t}{6}\right)\leq 2\exp\left(-c\cdot n\cdot\min\left(\frac{t^2}{\tau^4},\frac{t}{\tau^2}\right)\right),$$

(55)

for some universal constant $c$. As a result,

$$\mathbb{P}\left(B_t\right)\leq 2\exp\left(-c\cdot n\cdot\min\left(\frac{t^2}{\tau^4},\frac{t}{\tau^2}\right)+dK\log\frac{3r}{\epsilon}+dK\log 12\right).$$

(56)

Thus as long as

$$t > C\cdot\max\left\{\sqrt{\frac{\tau^4\left(dK\log\frac{36r}{\epsilon}+\log\frac{4}{\delta}\right)}{n}},\frac{\tau^2\left(dK\log\frac{36r}{\epsilon}+\log\frac{4}{\delta}\right)}{n}\right\}$$

(57)

for some large enough constant $C$, we have $\mathbb{P}\left(B_t\right)\leq\frac{\delta}{2}$.

2. **Upper bound $\mathbb{P}\left(A_t\right)$ and $\mathbb{P}\left(C_t\right)$.** These two events will be bounded in a similar way. Let $J_\star$ satisfy

$$\mathbb{E}\left[\sup_{\boldsymbol{W}\neq\boldsymbol{W}'\in\mathbb{B}(\boldsymbol{W}^\star,r)}\frac{\|\nabla^2\ell\left(\boldsymbol{W},\boldsymbol{x}\right)-\nabla^2\ell\left(\boldsymbol{W}',\boldsymbol{x}\right)\|}{\|\boldsymbol{W}-\boldsymbol{W}'\|_{\mathrm{F}}}\right]\leq J_\star.$$

(58)

Let us look at the deterministic event $C_t$ first. Since

$$\sup_{\boldsymbol{W}\in\mathbb{B}(\boldsymbol{W}^\star,r)}\|\mathbb{E}\left[\nabla^2\ell\left(\boldsymbol{W}_{j(\boldsymbol{W})};\boldsymbol{x}\right)\right]-\mathbb{E}\left[\nabla^2\ell\left(\boldsymbol{W};\boldsymbol{x}\right)\right]\|$$

$$\leq\sup_{\boldsymbol{W}\in\mathbb{B}(\boldsymbol{W}^\star,r)}\frac{\|\mathbb{E}\left[\nabla^2\ell\left(\boldsymbol{W}_{j(\boldsymbol{W})};\boldsymbol{x}\right)\right]-\mathbb{E}\left[\nabla^2\ell\left(\boldsymbol{W};\boldsymbol{x}\right)\right]\|}{\|\boldsymbol{W}-\boldsymbol{W}_{j(\boldsymbol{W})}\|_{\mathrm{F}}}\cdot\sup_{\boldsymbol{W}\in\mathbb{B}(\boldsymbol{W}^\star,r)}\|\boldsymbol{W}-\boldsymbol{W}_{j(\boldsymbol{W})}\|_{\mathrm{F}}$$

$$\leq J_\star\cdot\epsilon.$$

(59)

Therefore, $C_t$ holds as long as

$$t\geq 3J_\star\cdot\epsilon.$$

(60)

We can bound the event $A_t$ as below.

$$\mathbb{P}\left(A_t\right)=\mathbb{P}\left(\sup_{\boldsymbol{W}\in\mathbb{B}(\boldsymbol{W}^\star,r)}\frac{1}{n}\left\|\sum_{i=1}^{n}\left[\nabla^2\ell\left(\boldsymbol{W};\boldsymbol{x}_i\right)-\nabla^2\ell\left(\boldsymbol{W}_{j(\boldsymbol{W})};\boldsymbol{x}_i\right)\right]\right\|\geq\frac{t}{3}\right)$$

$$\leq\frac{3}{t}\mathbb{E}\left[\sup_{\boldsymbol{W}\in\mathbb{B}(\boldsymbol{W}^\star,r)}\left\|\frac{1}{n}\sum_{i=1}^{n}\left[\nabla^2\ell\left(\boldsymbol{W};\boldsymbol{x}_i\right)-\nabla^2\ell\left(\boldsymbol{W}_{j(\boldsymbol{W})};\boldsymbol{x}_i\right)\right]\right\|\right]$$

(61)

$$\leq\frac{3}{t}\mathbb{E}\left[\sup_{\boldsymbol{W}\in\mathbb{B}(\boldsymbol{W}^\star,r)}\left\|\nabla^2\ell\left(\boldsymbol{W};\boldsymbol{x}_i\right)-\nabla^2\ell\left(\boldsymbol{W}_{j(\boldsymbol{W})};\boldsymbol{x}_i\right)\right\|\right]$$

$$\leq\frac{3}{t}\mathbb{E}\left[\sup_{\boldsymbol{W}\in\mathbb{B}(\boldsymbol{W}^\star,r)}\frac{\|\nabla^2\ell\left(\boldsymbol{W};\boldsymbol{x}_i\right)-\nabla^2\ell\left(\boldsymbol{W}_{j(\boldsymbol{W})};\boldsymbol{x}_i\right)\|}{\|\boldsymbol{W}-\boldsymbol{W}_{j(\boldsymbol{W})}\|_{\mathrm{F}}}\right]\cdot\sup_{\boldsymbol{W}\in\mathbb{B}(\boldsymbol{W}^\star,r)}\|\boldsymbol{W}-\boldsymbol{W}_{j(\boldsymbol{W})}\|_{\mathrm{F}}$$

$$\leq\frac{3J_\star\epsilon}{t}$$

(62)

where (61) follows from the Markov inequality. Thus, taking

$$t\geq\frac{6\epsilon J_\star}{\delta}$$

(63)

ensures that $\mathbb{P}\left(A_t\right)\leq\frac{\delta}{2}$. It now boils down to control the quantity $J_\star$, which we have the following lemma, whose proof is in Appendix E.3.

**Lemma 8.** *There exists some constant $C$ such that*

$$\mathbb{E}\left[\sup_{\boldsymbol{W}\neq\boldsymbol{W}'\in\mathbb{B}(\boldsymbol{W}^\star,r)}\frac{\|\nabla^2\ell(\boldsymbol{W},\boldsymbol{x})-\nabla^2\ell(\boldsymbol{W}',\boldsymbol{x})\|}{\|\boldsymbol{W}-\boldsymbol{W}'\|_{\mathrm{F}}}\right]\leq C\cdot d\sqrt{K}\equiv J_\star. \tag{64}$$

3. **Final step.** Let $\epsilon=\frac{\delta\tau^2}{6J_\star\cdot ndK}$, $\delta=d^{-10}$ plugging into (57) we need

$$t>\tau^2\cdot\max\left\{\frac{1}{ndK},C\cdot\sqrt{\frac{\left(dK\log(36rnd^{11}K)+\log\frac{4}{\delta}\right)}{n}},\frac{\left(dK\log(36rnd^{11}K)+\log\frac{4}{\delta}\right)}{n}\right\}. \tag{65}$$

Since the middle term can be expressed as

$$\frac{dK\log(36rnd^{11}K)+10\log d}{n}\leq\frac{dK\log n}{n}+\frac{dK\log 36r}{n}+\frac{11dK\log dK}{n}+\frac{10\log d}{n}, \tag{66}$$

when $n\geq C\cdot dK\log dK$ for some large enough constant $C$, the first term, $dK\log n$ dominants and is on the order of $dK\log dK$. Moreover, it decreases as $n$ increases when $n\geq 3$. Thus we can set

$$t\geq\tau^2\sqrt{\frac{\left(dK\log(36rnd^{11}K)+\log\frac{4}{\delta}\right)}{n}} \tag{67}$$

which holds as $t\geq C'\cdot\tau^2\sqrt{\frac{dK\log n}{n}}$ for some constant $C'$.

By setting $t:=C\tau^2\sqrt{\frac{dK\log n}{n}}$ for sufficiently large $C$, as long as $n\geq C'\cdot dK\log dK$,

$$\mathbb{P}\left(\sup_{\boldsymbol{W}\in\mathbb{B}(\boldsymbol{W}^\star,r)}\|\nabla^2 f_n(\boldsymbol{W})-\nabla^2 f(\boldsymbol{W})\|\geq C\tau^2\sqrt{\frac{dK\log n}{n}}\right)\leq d^{-10}. \tag{68}$$

$\square$

### D.5 PROOF OF LEMMA 4

*Proof.* We nedd the following Lemma for the proof. Lemma 9

**Lemma 9.** *Assume $\boldsymbol{x}\sim\mathcal{N}(\boldsymbol{0},\boldsymbol{I})$. Let $\boldsymbol{u}$ be a fixed unit norm vector $\boldsymbol{u}=\left[\boldsymbol{u}_1^\top,\cdots,\boldsymbol{u}_K^\top\right]\in\mathbb{R}^{dK}$ with $\|\boldsymbol{u}\|_2=1$, the following*

$$\|\boldsymbol{u}^\top\nabla\ell(\boldsymbol{W};\boldsymbol{x})\|_{\psi_2}\leq\sqrt{K},$$

*hold.*

By a similar argument (details omitted) as the proof of Lemma 3, and applies Lemma 9, we can get the following concentration inequality:

$$\sup_{\boldsymbol{W}\in\mathbb{B}(\boldsymbol{W}^\star,r)}\|\nabla f_n(\boldsymbol{W})-\nabla f(\boldsymbol{W})\|_2\leq C\cdot\sqrt{\frac{d\sqrt{K}\log n}{n}}, \tag{69}$$

holds with probability at least $1-d^{-10}$, as long as the sample size $n\geq C\cdot dK\log(dK)$. $\square$

## E PROOF OF AUXILIARY LEMMAS

### E.1 PROOF OF LEMMA 5

*Proof.* We can rewrite the left-hand side as

$$\mathbb{E}\left[\left(\frac{1}{K^2}\sum_{i=1}^{K}\sum_{j=1}^{K}\phi\left(\boldsymbol{w}_i^\top\boldsymbol{x}\right)\left(1-\phi\left(\boldsymbol{w}_j^\top\boldsymbol{x}\right)\right)\right)^{-z}\right], \tag{70}$$

which is upper bounded by $\mathbb{E}\left[\frac{1}{K^2}\sum_{i=1}^{K}\sum_{j=1}^{K}\left(\phi\left(\boldsymbol{w}_i^\top\boldsymbol{x}\right)\left(1-\phi\left(\boldsymbol{w}_j^\top\boldsymbol{x}\right)\right)\right)^{-z}\right]$, since $f\left(x\right)=x^{-z}$ is convex for $x>0$ and $z\geq 1$. And apply Cauchy-Schwarz inequality we can have

$$\mathbb{E}\left[\left(\phi\left(\boldsymbol{w}_i^\top\boldsymbol{x}\right)\left(1-\phi\left(\boldsymbol{w}_j^\top\boldsymbol{x}\right)\right)\right)^{-z}\right]\leq\sqrt{\mathbb{E}\left[\phi\left(\boldsymbol{w}_i^\top\boldsymbol{x}\right)^{-2z}\right]}\cdot\sqrt{\mathbb{E}\left[\left(1-\phi\left(\boldsymbol{w}_j^\top\boldsymbol{x}\right)\right)^{-2z}\right]}. \quad (71)$$

Further since $\frac{1}{\phi(x)}=1+e^{-x}$, $\frac{1}{1-\phi(x)}=1+e^x$ and $g=\boldsymbol{w}_i^\top\boldsymbol{x}\sim\mathcal{N}\left(0,\sigma_i^2=\|\boldsymbol{w}_i\|_2^2\right)$, then we can exactly calculate the two terms in the above equation, i.e.,

$$\mathbb{E}\left[\phi\left(g\right)^{-2z}\right]=\mathbb{E}\left[\left(1+e^{-g}\right)^{2z}\right]=\mathbb{E}\left[\sum_{l=0}^{2z}\binom{2z}{l}e^{-lg}\right]=\sum_{l=0}^{2z}\binom{2z}{l}e^{\left(\frac{\sigma_i^2 l^2}{2}\right)}, \quad (72)$$

and in the same way,

$$\mathbb{E}\left[\left(1-\phi\left(g\right)\right)^{-2z}\right]=\mathbb{E}\left[\left(1+e^g\right)^{2z}\right]=\sum_{l=0}^{2z}\binom{2z}{l}e^{\left(\frac{\sigma_i^2 l^2}{2}\right)}, \quad (73)$$

since $g$ is a Gaussian random which is a symmetric random variable. Plugging this back into (71) we can conclude that for $t=\max\left(\|\boldsymbol{w}_1\|_2,\cdots,\|\boldsymbol{w}_K\|_2\right)$ and $p\geq 1$,

$$\mathbb{E}\left[\left(\frac{1}{\frac{1}{K}\sum_{i=1}^{K}\phi\left(\boldsymbol{w}_i^\top\boldsymbol{x}\right)\left(1-\frac{1}{K}\sum_{i=1}^{K}\phi\left(\boldsymbol{w}_i^\top\boldsymbol{x}\right)\right)}\right)^p\right]\leq C\cdot e^{t^2}, \quad (74)$$

holds. $\qquad\square$

### E.2 Proof of Lemma 7

*Proof.* The sub-exponential norm of $G_i$ can be bounded as

$$\|G_i\|_{\psi_1}\leq\|\left\langle\boldsymbol{u},\nabla^2\ell\left(\boldsymbol{W};z\right)\boldsymbol{u}\right\rangle\|_{\psi_1}+\|\nabla^2 f\left(\boldsymbol{W};z\right)\|,$$

where $\|\nabla^2 f\left(\boldsymbol{W};z\right)\|$ is upper bounded by $\frac{C}{K}$ according to lemma 2, and denote the $(j,l)$-th block of $\nabla^2\ell\left(\boldsymbol{W};z\right)$ as $\alpha_{j,l}\cdot\boldsymbol{x}\boldsymbol{x}^\top$, we can write

$$\|\left\langle\boldsymbol{u},\nabla^2\ell\left(\boldsymbol{W};z\right)\boldsymbol{u}\right\rangle\|_{\psi_1}\leq\sum_{l=1}^{K}\sum_{j=1}^{K}\|\alpha_{j,l}\cdot\boldsymbol{u}_j^\top\boldsymbol{x}\boldsymbol{x}^\top\boldsymbol{u}_l\|_{\psi_1}$$

$$\leq\sum_{l=1}^{K}\sum_{j=1}^{K}\sup_{t\geq 1}\quad t^{-1}\left(\mathbb{E}\left|\alpha_{j,l}\cdot\boldsymbol{u}_j^\top\boldsymbol{x}\boldsymbol{x}^\top\boldsymbol{u}_l\right|^t\right)^{\frac{1}{t}}. \quad (75)$$

Note that

- for $j\neq l$

$$\alpha_{j,l}=\frac{1}{K^2}\frac{\phi'\left(\boldsymbol{w}_j^\top\boldsymbol{x}\right)\phi'\left(\boldsymbol{w}_l^\top\boldsymbol{x}\right)\cdot\left(p\left(\boldsymbol{W}\right)^2+y-2y\cdot p\left(\boldsymbol{W}\right)\right)}{\left(p\left(\boldsymbol{W}\right)\left(1-p\left(\boldsymbol{W}\right)\right)\right)^2}, \quad (76)$$

further since

$$p\left(\boldsymbol{W}\right)^2+y-2y\cdot p\left(\boldsymbol{W}\right)\leq\begin{cases}p\left(\boldsymbol{W}\right)^2 & p\left(\boldsymbol{W}\right)>\frac{1}{2}\\\left(1-p\left(\boldsymbol{W}\right)\right)^2 & p\left(\boldsymbol{W}\right)\leq\frac{1}{2}\end{cases}, \quad (77)$$

then

$$|\alpha_{j,l}|\leq\begin{cases}\frac{1}{K^2}\frac{\phi'\left(\boldsymbol{w}_j^\top\boldsymbol{x}\right)\phi'\left(\boldsymbol{w}_l^\top\boldsymbol{x}\right)}{\left(1-p\left(\boldsymbol{W}\right)\right)^2} & p\left(\boldsymbol{W}\right)>\frac{1}{2}\\\frac{1}{K^2}\frac{\phi'\left(\boldsymbol{w}_j^\top\boldsymbol{x}\right)\phi'\left(\boldsymbol{w}_l^\top\boldsymbol{x}\right)}{p\left(\boldsymbol{W}\right)^2} & p\left(\boldsymbol{W}\right)\leq\frac{1}{2}\end{cases}, \quad (78)$$

moreover,

$$p\left(\boldsymbol{W}\right)^2 = \left(\frac{1}{K}\sum_{i=1}^{K}\phi\left(\boldsymbol{w}_i^\top \boldsymbol{x}\right)\right)^2 \geq \frac{1}{K^2}\phi\left(\boldsymbol{w}_j^\top \boldsymbol{x}\right)\phi\left(\boldsymbol{w}_l^\top \boldsymbol{x}\right)$$

$$\left(1 - p\left(\boldsymbol{W}\right)\right)^2 = \left(1 - \frac{1}{K}\sum_{i=1}^{K}\phi\left(\boldsymbol{w}_i^\top \boldsymbol{x}\right)\right)^2 \geq \frac{1}{K^2}\left(1 - \phi\left(\boldsymbol{w}_j^\top \boldsymbol{x}\right)\right)\left(1 - \phi\left(\boldsymbol{w}_l^\top \boldsymbol{x}\right)\right)$$

and recall that $\phi\left(x\right)\left(1 - \phi\left(x\right)\right) = \phi'\left(x\right)$, together we can obtain

$$|\alpha_{j,l}| \leq \begin{cases} \phi\left(\boldsymbol{w}_j^\top \boldsymbol{x}\right)\phi\left(\boldsymbol{w}_l^\top \boldsymbol{x}\right) \leq 1 & p\left(\boldsymbol{W}\right) > \frac{1}{2} \\ \left(1 - \phi\left(\boldsymbol{w}_j^\top \boldsymbol{x}\right)\right)\cdot\left(1 - \phi\left(\boldsymbol{w}_l^\top \boldsymbol{x}\right)\right) \leq 1 & p\left(\boldsymbol{W}\right) \leq \frac{1}{2} \end{cases}. \tag{79}$$

- for $j = l$:

$$|\alpha_{j,j}| \leq \left|\frac{1}{K^2}\frac{\phi'\left(\boldsymbol{w}_j^\top \boldsymbol{x}\right)\phi'\left(\boldsymbol{w}_l^\top \boldsymbol{x}\right)\cdot\left(p\left(\boldsymbol{W}\right)^2 + y - 2y\cdot p\left(\boldsymbol{W}\right)\right)}{\left(p\left(\boldsymbol{W}\right)\left(1 - p\left(\boldsymbol{W}\right)\right)\right)^2}\right|$$

$$+ \left|\frac{1}{K}\frac{\phi''\left(\boldsymbol{w}_j^\top \boldsymbol{x}\right)\left(y - p\left(\boldsymbol{W}\right)\right)}{p\left(\boldsymbol{W}\right)\left(1 - p\left(\boldsymbol{W}\right)\right)}\right|, \tag{80}$$

the first term is upper bounded by a constant, and for the second term

$$\left|\frac{\phi''\left(\boldsymbol{w}_j^\top \boldsymbol{x}\right)\left(1_{\{y=1\}} - p\left(\boldsymbol{W}\right)\right)}{p\left(\boldsymbol{W}\right)\left(1 - p\left(\boldsymbol{W}\right)\right)}\right| \leq \begin{cases} \frac{\phi''\left(\boldsymbol{w}_j^\top \boldsymbol{x}\right)}{\left(1 - p\left(\boldsymbol{W}\right)\right)} \leq K & y = 0 \\ \frac{\phi''\left(\boldsymbol{w}_j^\top \boldsymbol{x}\right)}{p\left(\boldsymbol{W}\right)} \leq K & y = 1 \end{cases}, \tag{81}$$

where we have used the fact that the second derivative is $\phi''\left(x\right) = \phi\left(x\right)\left(1 - \phi\left(x\right)\right)\left(1 - 2\phi\left(x\right)\right)$, the absolute value of which can be upper bounded by $\phi\left(x\right)$ or $1 - \phi\left(x\right)$. Thus we can show that

$$|\alpha_{j,j}| \leq C. \tag{82}$$

Finally, we conclude that

$$|\alpha_{j,l}| \leq C, \tag{83}$$

holds for all $j, l$. And

$$\|\langle \boldsymbol{u}, \nabla^2 \ell\left(\boldsymbol{W}; z\right)\boldsymbol{u}\rangle\|_{\psi_1} \leq C \cdot \sum_{l=1}^{K}\sum_{j=1}^{K}\sup_{t\geq 1} \quad t^{-1}\mathbb{E}\left[|\left(\boldsymbol{u}_j^\top \boldsymbol{x}\boldsymbol{x}^\top \boldsymbol{u}_l\right)|^t\right]^{\frac{1}{t}} \tag{84}$$

$$\leq C \cdot \sum_{l=1}^{K}\sum_{j=1}^{K}\sup_{t\geq 1} \quad t^{-1}\left(\sqrt{\mathbb{E}\left[\left(\boldsymbol{u}_j^\top \boldsymbol{x}\right)^{2t}\right]}\cdot\sqrt{\mathbb{E}\left[\left(\boldsymbol{u}_l^\top \boldsymbol{x}\right)^{2t}\right]}\right)^{\frac{1}{t}} \tag{85}$$

$$\leq C \cdot \sum_{l=1}^{K}\sum_{j=1}^{K}\|\boldsymbol{u}_j\|_2\|\boldsymbol{u}_l\|_2 \cdot \sup_{p\geq 1} \quad t^{-1}\left((2t-1)!!\right)^{\frac{1}{t}} \tag{86}$$

$$\leq C \sum_{l=1}^{K}\sum_{j=1}^{K}\|\boldsymbol{u}_j\|_2\|\boldsymbol{u}_l\|_2 \tag{87}$$

$$\leq C \sum_{l=1}^{K}\sum_{j=1}^{K}\frac{\|\boldsymbol{u}_j\|_2^2 + \|\boldsymbol{u}_l\|_2^2}{2} \tag{88}$$

$$\leq C :\equiv \tau^2 \tag{89}$$

Thus we can conclude that

$$\|G_i\|_{\psi_1} \leq C :\equiv \tau^2$$

$\square$

### E.3    PROOF OF LEMMA 8

*Proof.* As noted before, we can write the $(j, l)$-th block of $\nabla^2 \ell (\boldsymbol{W}; \boldsymbol{z})$ as $g_{j,l} (\boldsymbol{W}) \boldsymbol{x} \boldsymbol{x}^\top$, where

$$g_{j,l} (\boldsymbol{W}) = \frac{\xi_{j,l} (\boldsymbol{W})}{\left( p (\boldsymbol{W}) (1 - p (\boldsymbol{W})) \right)^2}, \tag{90}$$

then we can obtain the following bound,

$$\| \nabla^2 \ell (\boldsymbol{W}; z) - \nabla^2 \ell (\boldsymbol{W}'; z) \| \leq \sum_{j=1}^K \sum_{l=1}^K |g_{j,l} (\boldsymbol{W}) - g_{j,l} (\boldsymbol{W}')| \cdot \| \boldsymbol{x} \boldsymbol{x}^\top \|. \tag{91}$$

Using the same method as shown in the proof of Lemma 1, we can upper bound $|g_{j,l} (\boldsymbol{W}) - g_{j,l} (\boldsymbol{W}')|$ as

$$|g_{j,l} (\boldsymbol{W}) - g_{j,l} (\boldsymbol{W}')| \leq \frac{1}{K^2} \frac{1}{\left( p \left( \widetilde{\boldsymbol{W}} \right) \left( 1 - p \left( \widetilde{\boldsymbol{W}} \right) \right) \right)^6} \cdot \| \boldsymbol{x} \|_2 \cdot \sqrt{K} \cdot \| \boldsymbol{W} - \boldsymbol{W}' \|_F \tag{92}$$

where $\widetilde{\boldsymbol{W}} = \eta \boldsymbol{W} + (1 - \eta) \boldsymbol{W}'$ for $\eta \in (0, 1)$. And thus, when $\| \boldsymbol{W} - \boldsymbol{W}' \|_F \leq 0.7$ we have

$$\mathbb{E} \left[ \sup_{\boldsymbol{W} \neq \boldsymbol{W}'} \frac{\| \nabla^2 \ell (\boldsymbol{W}) - \nabla^2 \ell (\boldsymbol{W}') \|}{\| \boldsymbol{W} - \boldsymbol{W}' \|_F} \right] \leq \frac{C}{K^{\frac{3}{2}}} \cdot K^2 \cdot \mathbb{E} \left[ \frac{1}{\left( p \left( \widetilde{\boldsymbol{W}} \right) \left( 1 - p \left( \widetilde{\boldsymbol{W}} \right) \right) \right)^6} \cdot \| \boldsymbol{x} \|_2 \cdot \| \boldsymbol{x} \boldsymbol{x}^\top \| \right]$$

$$\leq C \cdot d \sqrt{K} \tag{93}$$

Thus we only need to set $J^\star \geq C \cdot d \sqrt{K}$ for some large enough $C$.    $\square$

### E.4    PROOF OF LEMMA 9

*Proof.* By definition, we have

$$\langle \nabla \ell (\boldsymbol{W}), \boldsymbol{u} \rangle = \sum_{k=1}^K \left\langle \frac{\partial \ell (\boldsymbol{W})}{\partial \boldsymbol{w}_k}, \boldsymbol{u}_k \right\rangle$$

$$= \frac{1}{K} \sum_{k=1}^K \left( \frac{\left( y - \frac{1}{K} \sum_{i=1}^K \phi \left( \boldsymbol{w}_i^\top \boldsymbol{x} \right) \right) \cdot \phi' \left( \boldsymbol{w}_k^\top \boldsymbol{x} \right)}{\frac{1}{K} \sum_{i=1}^K \phi \left( \boldsymbol{w}_i^\top \boldsymbol{x} \right) \left( 1 - \frac{1}{K} \sum_{i=1}^K \phi \left( \boldsymbol{w}_i^\top \boldsymbol{x} \right) \right)} \right) \left( \boldsymbol{u}_k^\top \boldsymbol{x} \right),$$

and then we can upper bound the sub-gaussian norm as

$$\| \langle \nabla \ell (\boldsymbol{W}), \boldsymbol{u} \rangle \|_{\psi_2} \leq \begin{cases} \frac{1}{K} \sum_{k=1}^K \left\| \frac{\phi' \left( \boldsymbol{w}_k^\top \boldsymbol{x} \right) \cdot \boldsymbol{u}_k^\top \boldsymbol{x}}{\left( 1 - \frac{1}{K} \sum_{i=1}^K \phi \left( \boldsymbol{w}_i^\top \boldsymbol{x} \right) \right)} \boldsymbol{u}_k^\top \boldsymbol{x} \right\|_{\psi_2} \leq \sum_{k=1}^K \| \boldsymbol{u}_k^\top \boldsymbol{x} \|_{\psi_2} & y = 0 \\ \frac{1}{K} \sum_{k=1}^K \left\| \frac{\phi' \left( \boldsymbol{w}_k^\top \boldsymbol{x} \right) \cdot \boldsymbol{u}_k^\top \boldsymbol{x}}{\frac{1}{K} \sum_{i=1}^K \phi \left( \boldsymbol{w}_i^\top \boldsymbol{x} \right)} \boldsymbol{u}_k^\top \boldsymbol{x} \right\|_{\psi_2} \leq \sum_{k=1}^K \| \boldsymbol{u}_k^\top \boldsymbol{x} \|_{\psi_2} & y = 1 \end{cases}.$$

Thus we can have

$$\| \langle \nabla \ell (\boldsymbol{W}), \boldsymbol{u} \rangle \|_{\psi_2} \leq \sum_{k=1}^K \| \boldsymbol{u}_k \|_2 \sqrt{K}, \tag{94}$$

and conclude that the directional gradient is $\sqrt{K}$-sub-Gaussian.    $\square$

