# OpenReview forum: "Guaranteed Recovery of One-Hidden-Layer Neural Networks via Cross Entropy"
_ICLR.cc/2019/Conference_

### Official Review · AnonReviewer2 · 2018-10-31

**Rating:** 5
**Confidence:** 4

**Review:**

Paper Summary:
This paper studies the problem of recovering a true underlying neural network (assuming there exists one) with cross-entropy loss. This paper shows if the input is standard Gaussian, within a small ball around the ground truth, the objective function is strongly convex and smooth if there is a sufficiently large number of samples. Furthermore, the global minimizer is actually the true neural network. This geometric analysis implies applying gradient descent within this neighborhood, one can recover the underlying neural network. This paper also proposed a provable method based on spectral learning to find a good initialization point. Lastly, this paper also provides some simulation studies.

Comments:
This paper closely follows a recent line of work on recovering a neural network under Gaussian input assumption. While studying cross-entropy loss is interesting, the analysis techniques in this paper are very similar to Zhong et al. 2017, so this paper is incremental. I believe studying the global convergence of the gradient descent or relaxing the Gaussian input assumption is more interesting.

---

### Official Review · AnonReviewer1 · 2018-11-03
**Incremental work, not strong enough**

**Rating:** 4
**Confidence:** 4

**Review:**

This paper studies the problem of learning the parameter of one hidden layer neural network with sigmoid activation function based on the negative log likelihood loss. The authors consider the teacher network setting with Gaussian input, and show that gradient descent can recover the teacher network’s parameter up to certain statistical accuracy when the initialization is sufficiently close to the true parameter. The main contribution of this paper is that the authors consider the classification problem with negative log likelihood loss, and provide the local convergence result for gradient descent. However, based on the previous results in Mei et al., 2016 and Zhong et al., 2017, this work is incremental, and current results in this paper is not strong enough. To be more specific, the paper has the following weaknesses:

1.	The authors show the uniformly strongly convex and smooth property of the objective loss function which can get rid of the sample splitting procedure used in Zhong et al., 2017. However, the method for proving this uniform result has been previously used in Mei et al., 2016. And the extension to the negative log likelihood objective function is straightforward since the derivate and Hessian of the log likelihood function can be easily bounded given the sigmoid activation function.
2.	The authors employ a tensor initialization algorithm proposed by Zhong et al, 2017 to satisfy their initialization requirement. However, it seems like that the tensor initialization almost enables the recovery as it already lands on a point close to the ground truth, the role of GD is somehow not
that crucial. If the authors can prove the convergence of GD with random initialization, the results of this paper will be much stronger.
3.	The presentation of the current paper needs to be improved. The authors should distinguish \cite and \citep. There are some incomplete sentences in the current paper, such as in page 3, “Moreover, (Zhong et al., 2017b) shows…the ground truth From a technical perspective, our…”.

---

### Official Review · AnonReviewer3 · 2018-11-03
**Lack of practicality and theoretical depth.**

**Rating:** 3
**Confidence:** 4

**Review:**

The paper presents theoretical analysis for recovering one-hidden-layer neural networks using logistic loss function. I have the following major concerns:

(1.a) The paper does not mention identifiability at all. As has been known, neural networks with even only one hidden layer are not identifiable. The authors need to either prove the identifiability or cite existing references on the identifiability. Otherwise,  the parameter recovery does not make sense.

Example: The linear network takes f(x) = 1'Wx/k, where 1 is a vector with every entry equal to one. Then two models with parameters W and V are identical as long 1'W = 1'V.

(1.b) If the equivalent parameters are not isolated, the local strong convexity is impossible to hold. The authors need to carefully justify their claim.

(2) When using Sigmoid or Tanh activation functions, the output is bounded between [0,1] or [-1,+1]. This is unrealistic for logistic regression: The output of [0,1] means that the posterior probability has to be bounded between 1/2 and e/(1+e); The output of [-1,1] means that the posterior probability has to be bounded between 1/(1+e) and e/(1+e).

(3) The most challenging part of the logistic loss is the lack of curvature, when neural networks have large magnitude outputs. Since this paper assumes that the neural networks takes very small magnitude outputs, the extension from Zhong et al. 2017b to the logistic loss is very straightforward.

(4) Spectral initialization is very impractical. Nobody is using it in practice. The spectral initialization avoids the challenging global convergence analysis.

(5) Theorem 3 needs clarification. Please explicitly write the RHS of (7). The result would become meaningless, if under the scaling of Theorem 2, is the RHS of (7) smaller than RHS of (5).

I also have the following minor concerns on some unrealistic assumptions, but these concerns do not affect my rating. These assumptions have been widely used in many other papers, due to the lack of theoretical understanding of neural networks in the machine learning community.

(6)	The neural networks take independent Gaussian input.
(7)	The model is assumed to be correct.
(8)	Only gradient descent is considered.

---

### Meta-Review · Area_Chair1 · 2018-12-11
**ICLR 2019 decision**

**Confidence:** 4
**Recommendation:** Reject

**Metareview:**

This paper shows local convergence results for gradient descent on one hidden layer network with Gaussian inputs and sigmoid activations. Later it shows global convergence by using spectral initialization. All the reviewers agree that the results are similar to existing work in the literature with little novelty. There are also some concerns about the correctness of the statements expressed by some reviewers.